# Molecular determinants of the Ska-Ndc80 interaction and their influence on microtubule tracking and force-coupling

Pim J Huis in 't Veld[1†], Vladimir A Volkov[2†], Isabelle D Stender[1], Andrea Musacchio[1,3]*, Marileen Dogterom[2]*

[1]Department of Mechanistic Cell Biology, Max Planck Institute of Molecular Physiology, Dortmund, Germany; [2]Department of Bionanoscience, Faculty of Applied Sciences, Delft University of Technology, Delft, Netherlands; [3]Centre for Medical Biotechnology, Faculty of Biology, University Duisburg, Essen, Germany

**Abstract** Errorless chromosome segregation requires load-bearing attachments of the plus ends of spindle microtubules to chromosome structures named kinetochores. How these end-on kinetochore attachments are established following initial lateral contacts with the microtubule lattice is poorly understood. Two microtubule-binding complexes, the Ndc80 and Ska complexes, are important for efficient end-on coupling and may function as a unit in this process, but precise conditions for their interaction are unknown. Here, we report that the Ska-Ndc80 interaction is phosphorylation-dependent and does not require microtubules, applied force, or several previously identified functional determinants including the Ndc80-loop and the Ndc80-tail. Both the Ndc80-tail, which we reveal to be essential for microtubule end-tracking, and Ndc80-bound Ska stabilize microtubule ends in a stalled conformation. Modulation of force-coupling efficiency demonstrates that the duration of stalled microtubule disassembly predicts whether a microtubule is stabilized and rescued by the kinetochore, likely reflecting a structural transition of the microtubule end.

*For correspondence:
andrea.musacchio@mpi-dortmund.mpg.de (AM);
m.dogterom@tudelft.nl (MD)

†These authors contributed equally to this work

Competing interests: The authors declare that no competing interests exist.

## Introduction

Correct attachment of chromosomes to spindle microtubules during eukaryotic cell division allows daughter cells to inherit the appropriate complement of chromosomes from their mothers and is therefore essential for life. Macromolecular structures named kinetochores generate physical links between chromosomes and microtubules (*Musacchio and Desai, 2017*). Kinetochores are built on specialized chromosome loci known as centromeres, and consist of centromere-proximal and centromere-distal layers of interacting proteins, known as the inner and the outer kinetochore. Within the latter, the 10-subunit Knl1-Mis12-Ndc80 (KMN) network functions both as a microtubule-capturing interface and as a control hub for a cell cycle checkpoint (spindle assembly checkpoint, SAC) that halts cells in mitosis until the correct configuration of chromosomes on the mitotic spindle (bi-orientation) has been reached.

Upon entry into M-phase (mitosis or meiosis) and spindle assembly, chromosomes are often initially transported to the spindle poles, where the microtubule density is highest, and from there to the spindle's equatorial plane, forming lateral attachments to the microtubule lattice. CENP-E, a kinetochore-localized, microtubule-plus-end-directed motor plays an essential function in this process (*Bancroft et al., 2015*; *Barisic et al., 2014*; *Cai et al., 2009*; *Chakraborty et al., 2019*; *Kapoor et al., 2006*; *Kim et al., 2008*; *Kitajima et al., 2011*; *Magidson et al., 2011*; *Shrestha et al., 2017*; *Tanaka et al., 2005*). In a poorly understood process of 'end-conversion', kinetochores engage the microtubule-binding interface of the KMN network and transition from binding to the lattice to binding to the dynamic plus ends of the microtubules, which become

embedded into the kinetochore's outer plate (*Dong et al., 2007*; *Kuhn and Dumont, 2017*; *McIntosh et al., 2013*; *Wan et al., 2009*). These so-called end-on attachments persist during polymerization and depolymerization of the dynamic ends of microtubules, and couple pulling forces produced by depolymerizing microtubules to chromosome movement (*Akiyoshi et al., 2010*; *Grishchuk et al., 2005*; *Miller et al., 2016*; *Powers et al., 2009*; *Volkov et al., 2018*). Furthermore, kinetochores control the dynamics of the plus ends, likely by balancing the action of MCAK (kinesin-13, a microtubule depolymerase) and Kif18 (kinesin 8, a microtubule stabilizer) and possibly other plus end-associated proteins (*Auckland and McAinsh, 2015*; *Monda et al., 2017*).

The molecular underpinnings of end-on attachment and tracking by kinetochores, and hence of force-coupling, remain unclear. However, two protein complexes, the Ndc80 and Ska complexes, have emerged for a prominent involvement in this process (*Figure 1A*) (*Auckland and McAinsh, 2015*; *Monda and Cheeseman, 2018*). The Ndc80 complex is part of the KMN network, which is stably bound to kinetochores during mitosis (*Cheeseman and Desai, 2008*). The KMN is crucially required for end-on microtubule attachment, and interference with its function leads to severe defects in chromosome alignment and SAC abrogation (*Cheeseman et al., 2006*; *DeLuca et al., 2005*; *DeLuca et al., 2006*; *Kim and Yu, 2015*; *McCleland et al., 2003*). In both humans and *Saccharomyces cerevisiae*, the four subunits of Ndc80 (NDC80/HEC1, NUF2, SPC25, and SPC24) have high coiled-coil content and form a ~ 60 nm dumbbell structure in which highly elongated NDC80: NUF2 and SPC25:SPC24 sub-complexes meet in a tetramerization domain (*Figure 1B*) (*Ciferri et al., 2005*; *Ciferri et al., 2008*; *Huis in 't Veld et al., 2016*; *Valverde et al., 2016*; *Wei et al., 2005*). At one end of Ndc80, two closely interacting calponin-homology (CH) domains near the N-terminal ends of NDC80 and NUF2 form a globular structure that binds the microtubule. An ~80 residue basic tail preceding the NDC80 CH-domain (Ndc80-tail) has also been implicated in microtubule binding, and phosphorylation by Aurora kinase activity has been proposed to modulate electrostatic interactions with the negatively charged MT lattice (*Alushin et al., 2012*; *Cheerambathur et al., 2017*; *Cheeseman et al., 2002*; *Cheeseman et al., 2006*; *Ciferri et al., 2008*; *DeLuca et al., 2006*; *DeLuca et al., 2011*; *DeLuca et al., 2018*; *Guimaraes et al., 2008*; *Long et al., 2017*; *Miller et al., 2008*; *Shrestha et al., 2017*; *Tooley et al., 2011*; *Umbreit et al., 2012*; *Wei et al., 2007*; *Ye et al., 2015*; *Zaytsev et al., 2015*; *Zaytsev et al., 2014*).

At the opposite end of Ndc80, C-terminal RWD domains in SPC25 and SPC24 mediate interactions with other kinetochore subunits to dock Ndc80 complexes onto the rest of the kinetochore (*Musacchio and Desai, 2017*). The coiled-coils flanking the globular domains form an apparently rigid rod, with a distinctive hinge point coinciding with a ~ 38 residue insertion (residues 427–464 of human NDC80, *Figure 1B*), known as the Ndc80 loop (*Ciferri et al., 2008*; *Wei et al., 2005*). The Ndc80 loop was proposed to be a site of interaction for other microtubule-binding proteins, a feature essential for end-on attachment and coupling to microtubule dynamics, or a tension sensor (*Hsu and Toda, 2011*; *Maure et al., 2011*; *Schmidt et al., 2012*; *Varma et al., 2012*; *Wan et al., 2009*; *Zhang et al., 2012*).

The Ska complex is crucially required to stabilize kinetochore-microtubule attachment (*Auckland et al., 2017*; *Daum et al., 2009*; *Gaitanos et al., 2009*; *Hanisch et al., 2006*; *Ohta et al., 2010*; *Raaijmakers et al., 2009*; *Rines et al., 2008*; *Sauer et al., 2005*; *Theis et al., 2009*; *Welburn et al., 2009*). Its three subunits (SKA, SKA2, and SKA3) are paralogs that interact through N-terminal coiled-coil segments and can further oligomerize into a dimer of SKA1-3 trimers (*Figure 1C*) (*Helgeson et al., 2018*; *Jeyaprakash et al., 2012*; *Maciejowski et al., 2017*; *Schmidt et al., 2012*; *van Hooff et al., 2017*). Ska can target microtubules autonomously through a microtubule-binding winged-helix-like domain in the C-terminal region of SKA1 (*Abad et al., 2014*; *Schmidt et al., 2012*). Depending on the severity of depletion, ablation of Ska results either in a metaphase-like arrest with weak kinetochore fibres, reduced inter-kinetochore tension, and SAC activation, or in a more dramatic alignment defect similar in severity to that observed upon Ndc80 depletion, despite lack of evident kinetochore damage (*Daum et al., 2009*; *Gaitanos et al., 2009*; *Hanisch et al., 2006*; *Ohta et al., 2010*; *Raaijmakers et al., 2009*; *Rines et al., 2008*; *Sauer et al., 2005*; *Theis et al., 2009*; *Welburn et al., 2009*). However, while Ndc80 is required for the SAC response (*Kim and Yu, 2015*; *McCleland et al., 2003*), Ska is not and its ablation results in strong SAC activation, prolonged mitotic arrest, and frequent cell death in mitosis. Ska is not present in all eukaryotes, but an evolutionary distinct complex, Dam1, usually performs an analogous, complementary function in organisms devoid of Ska (*van Hooff et al., 2017*).

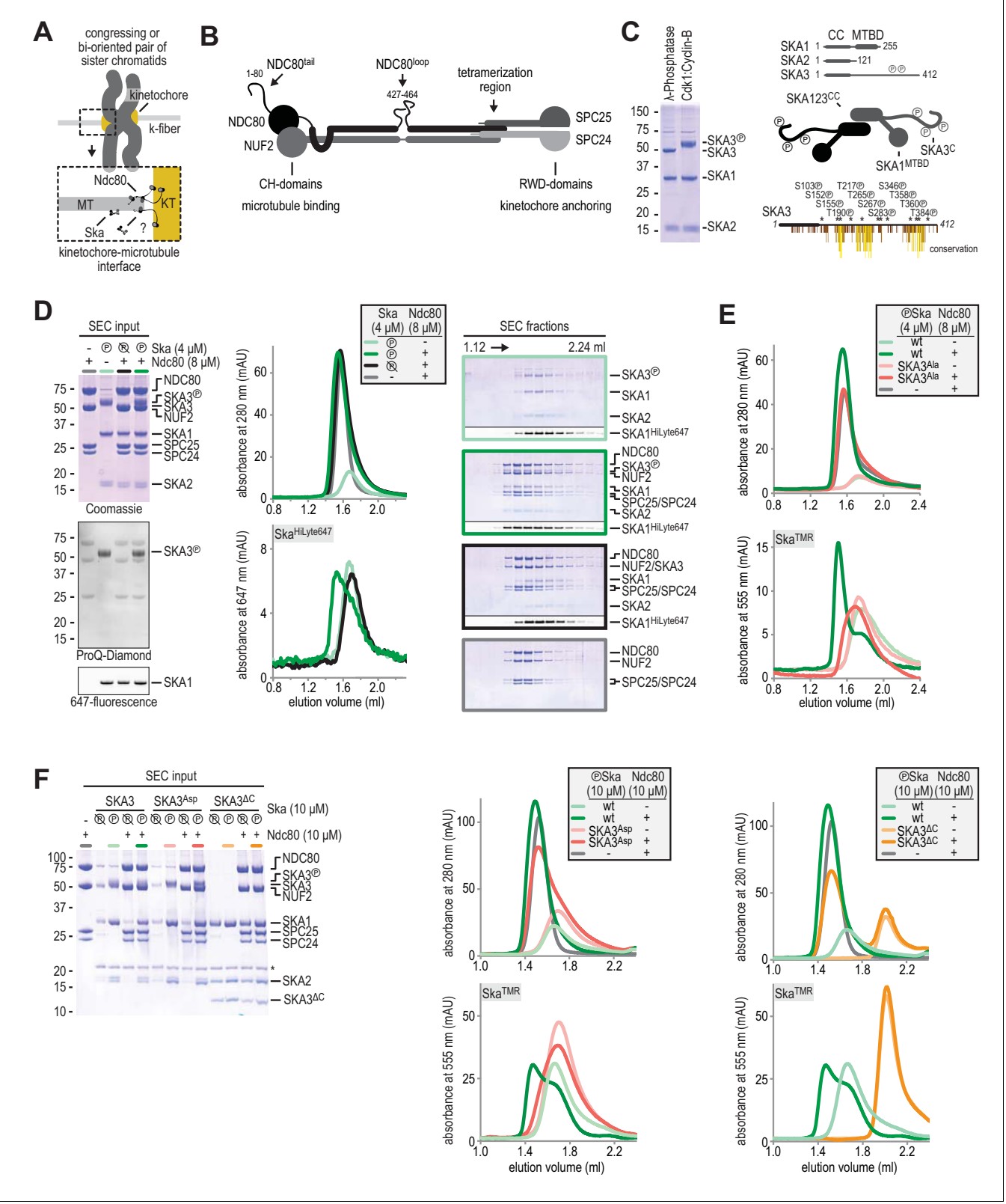

**Figure 1.** Formation of a Ska:Ndc80 complex upon SKA3 phosphorylation by CDK1. (**A**) Schematic representation of Ndc80 and Ska at the kinetochore-microtubule interface. (**B**) Overview of important regions in the Ndc80 complex. (**C**) The Ska complex. SKA1, SKA2, and SKA3 contain an N-terminal coiled coil (CC) region that mediates complex formation and dimerization. SKA1 contains a microtubule binding domain (MTBD). The largely unstructured C-terminal region of SKA3 is phosphorylated during mitosis. Multisite in vitro phosphorylation of purified Ska by CDK1:Cyclin-B altered the

*Figure 1 continued on next page*

*Figure 1 continued*

migration of SKA3 on SDS-PAGE. Identified phosphorylation sites and the conservation of SKA3 are shown. (D) Analysis of a Ska:Ndc80 mixture by size-exclusion chromatography (SEC) using a superose 6 increase 5/150 column shows that a stable complex is formed between Ska that is phosphorylated by CDK1:Cyclin-B and Ndc80. Elution of Ska from the column can be followed specifically through the fluorescently labelled SKA1. In-gel fluorescence of SKA1 in the SEC fractions analyzed by SDS-PAGE is also shown. (E) Phosphorylated Ska with SKA3$^{T358A/T360A}$ does not bind to Ndc80. Analysis of fractions is shown in *Figure 1—figure supplement 3*. (F) Ska without SKA3$^{104-412}$ as well as SKA3$^{T358D/T360D}$ does not interaction with Ndc80. A comparison with phosphatase-treated Ska on the input gel indicates the effective phosphorylation of the mutated SKA3. These chromatograms originate from one experiment and wild-type Ska (green) and Ndc80 (gray) are shown in both panels for comparison.

The online version of this article includes the following figure supplement(s) for figure 1:

**Figure supplement 1.** Hydrodynamic analysis of the SKA complex.

**Figure supplement 2.** Multiple species alignment of the region in SKA3 that has been implicated in the binding to Ndc80.

**Figure supplement 3.** Fractions of SEC experiments shown in *Figure 1E* and *Figure 1F* were analysed by Coomassie staining and in-gel fluorescence following SDS-PAGE.

Biochemical reconstitutions and various biophysical analyses have shed light into how Ndc80 and Ska contribute to microtubule binding, end-coupling, and load-bearing. Ndc80 interacts with the microtubule lattice by introducing a 'toe' of the NDC80 CH-domain into the interface between tubulin monomers and by additionally harnessing the N-terminal tail to increase binding affinity (*Alushin et al., 2012*; *Alushin et al., 2010*; *Ciferri et al., 2008*; *DeLuca and Musacchio, 2012*; *Sundin et al., 2011*; *Tooley et al., 2011*; *Wei et al., 2007*). Ndc80 has reduced binding affinity for features attributed to depolymerizing microtubule ends, such as curling protofilaments (*Powers et al., 2009*; *Welburn et al., 2009*). On the contrary, intrinsic features of the MT-binding domain of SKA1 allow it to interact preferentially with curved protofilaments that likely mimic a depolymerizing end (such as those obtained with certain microtubule poisons), compared to baseline affinity to the microtubule lattice (*Abad et al., 2014*; *Maciejowski et al., 2017*; *Schmidt et al., 2012*). Individual Ndc80 complexes are unable to track microtubule ends (*Lampert et al., 2010*; *Powers et al., 2009*; *Schmidt et al., 2012*; *Volkov et al., 2018*), suggesting that Ndc80 lacks intrinsic microtubule end-tracking properties, but it acquires the end-tracking activity in the context of multimerization (*Powers et al., 2009*; *Volkov et al., 2018*). Ska, on the other hand, can track depolymerizing and polymerizing plus ends as a dimer and possibly even as a monomer (*Helgeson et al., 2018*; *Monda et al., 2017*; *Schmidt et al., 2012*). Finally, both Ndc80 and Ska, each on their own, can form load-bearing attachments to microtubules when sparsely distributed on the surface of a bead trapped in an optical tweezer (*Helgeson et al., 2018*; *Powers et al., 2009*).

Thus, Ndc80 and Ska have, each in their own right, features expected of an end-coupler. Furthermore, these complexes may physically interact and bind microtubules in cooperation. First, Ska critically requires Ndc80 for kinetochore recruitment and its kinetochore levels increase after Ndc80-mediated end-on attachment (*Chan et al., 2012*; *Gaitanos et al., 2009*; *Hanisch et al., 2006*; *Raaijmakers et al., 2009*; *Welburn et al., 2009*; *Zhang et al., 2017*). Second, Ska complexes promote the tracking by Ndc80 of depolymerizing microtubule ends in vitro, and appear to increase the survival probability of attachments in vitro, both in the absence and in the presence of load (*Helgeson et al., 2018*; *Powers et al., 2009*; *Schmidt et al., 2012*). Importantly, Ska loads on kinetochores that are already bound to microtubules, and its kinetochore localization appears to be negatively regulated by Aurora kinase activity (*Chan et al., 2012*; *Hanisch et al., 2006*; *Redli et al., 2016*; *Schmidt et al., 2012*; *Sivakumar and Gorbsky, 2017*). Collectively, these observations have raised the interesting perspective that the interaction of Ska and Ndc80 may be directly regulated by force (*Cheerambathur et al., 2017*; *Helgeson et al., 2018*).

In the absence of assays exposing direct binding of Ska and Ndc80, previous studies have focused on effects on kinetochore recruitment after mutational perturbation of the two complexes, or on effects caused by combining Ska and Ndc80 on microtubules. For instance, the Ndc80 loop was identified as a crucial enabler of Ska binding (*Zhang et al., 2012*; *Zhang et al., 2017*). However, observations that Ska may interact on microtubules with Ndc80$^{bonsai}$, an engineered Ndc80 that lacks the loop region (*Ciferri et al., 2008*), seems inconsistent with this requirement (*Janczyk et al., 2017*). In another study, the Ndc80 N-terminal tail was shown to regulate the localization of Ska to kinetochores (*Cheerambathur et al., 2017*). In a key recent study, the C-terminal disordered region of SKA3 was shown to be sufficient for a direct interaction with an NDC80:NUF2 sub-complex after

phosphorylation in vitro by the Cdk1:Cyclin B kinase complex (*Zhang et al., 2017*). This observation requires further scrutiny, however, because other regions of Ska, such as a 'bridge' region (residues 92–132) of SKA1, are required for kinetochore recruitment of Ska (*Abad et al., 2014*). How phosphorylation regulates the interaction of Ska with kinetochores, and specifically with Ndc80, thus remains an important and unresolved question, not least because recombinant Ska and Ndc80 seem to interact on microtubules in the absence of phosphorylation (*Helgeson et al., 2018*; *Janczyk et al., 2017*; *Schmidt et al., 2012*).

These fragmented and contradictory views may reflect experimental conditions that fall short of capturing crucial properties (e.g. composition, geometry, force-induced conformational changes) of real kinetochore-microtubule attachment sites. Here, we identify, for the first time, conditions for the physical interaction of homogeneous, full-length recombinant Ska and Ndc80 in the absence of microtubules or applied force, and discuss them in light of previous work. The role of Aurora B in Ska recruitment may be more complex than hitherto believed, because its kinase activity does not evidently alter the interaction of the two complexes in vitro. Experimenting with total internal reflection fluorescence (TIRF) microscopy and optical tweezers, we harnessed our reconstitution to dissect how the Ndc80:Ska interaction affects microtubule end-tracking and force-coupling. We demonstrate that Ska extends Ndc80-mediated stalls of microtubule depolymerisation, often at higher stall forces, and identify the duration of a force-induced stall as a potentially crucial parameter in determining whether an end-on bound microtubule will restore its growth. In contrast, phosphorylation of the Ndc80-tail by Aurora B kinase specifically weakened end-on Ndc80-microtubule attachments under force by shortening the stalls. Taken together, our results have important implications for understanding the molecular basis of kinetochore-microtubule attachment.

## Results

### Ska directly binds Ndc80 upon CDK1:Cyclin B phosphorylation of SKA3$^{T358/T360}$

We co-expressed human SKA1, SKA2, and SKA3 from a single baculovirus in insect cells and purified the resulting Ska complex using consecutive metal-affinity, ion-exchange, and size-exclusion chromatography (SEC). SEC-MALS (multiangle light scattering) and SV-AUC (sedimentation velocity-analytical ultracentrifugation) analyses identified recombinant Ska as a dimer (*Figure 1—figure supplement 1*), in line with previous reports (*Helgeson et al., 2018*; *Jeyaprakash et al., 2012*; *Maciejowski et al., 2017*; *Schmidt et al., 2012*).

SKA3 is strongly phosphorylated in mitosis, and at least three kinases, Aurora B, MPS1, and CDK1, have been implicated in its phosphorylation (*Chan et al., 2012*; *Gaitanos et al., 2009*; *Maciejowski et al., 2017*; *Theis et al., 2009*; *Zhang et al., 2017*). Among these kinases, CDK1 appears to have a prominent role, because its inhibition suppresses the mitotic phosphorylation of Ska (*Zhang et al., 2017*). Furthermore, CDK1 phosphorylation of a ~ 310 residue C-terminal extension of SKA3, predicted to be largely intrinsically disordered and unstructured, has been proposed to promote binding to Ndc80 (*Zhang et al., 2017*). To verify these results, we phosphorylated Ska in vitro with CDK1:Cyclin B. This resulted in a readily detectable shift of the SKA3 subunit (*Figure 1C*). By mass spectrometry, we identified 12 of the 14 CDK consensus sites in the C-terminal part of SKA3$^{102-412}$ (SKA3$^C$) as being phosphorylated (*Figure 1C* and *Supplementary file 1a-1b*). These included two threonine sites within a conserved TPTP$^{358-361}$ sequence whose phosphorylation was previously shown to be required for kinetochore recruitment of the Ska complex (*Zhang et al., 2017*) (*Figure 1—figure supplement 2*). To test if the phosphorylation of Ska impacts its binding to Ndc80, we mixed both purified complexes at low micromolar concentrations and assessed their interaction using SEC. Sortase-mediated replacement of the C-terminal polyhistidine tag on SKA1 with a fluorescent label permitted specific monitoring of Ska. This allowed us to demonstrate that phosphorylated Ska, when mixed with Ndc80, elutes earlier from a SEC column, indicative of complex formation (*Figure 1D*, light and dark green traces). Ska that had been treated with λ-phosphatase after CDK1:Cyclin B treatment did not form a complex with Ndc80, indicating that this interaction depends on the phosphorylation of Ska by CDK1:Cyclin B (*Figure 1D*, black traces).

Alanine substitution of Thr358 and Thr360 in SKA3 prevented the recruitment of Ska to the kinetochore in vivo (*Zhang et al., 2017*). Mutations T358A and T360A in SKA3 also prevented efficient

formation of the Ska:Ndc80 complex in vitro (*Figure 1E* and *Figure 1—figure supplement 3A*). Thus, phosphorylation by CDK1:Cyclin B at other SKA3 sites, revealed by the phosphorylation induced shift of SKA3 on SDS-PAGE, was not sufficient to mediate Ska3:Ndc80 complex formation. It has also been shown that two phospho-mimetic mutations, T358D and T360D, are sufficient to promote robust Ska kinetochore localization when twelve additional potential phosphorylation target sites were mutated to alanine (*Zhang et al., 2017*). However, neither unphosphorylated nor phosphorylated Ska containing the T358D and T360D mutations bound Ndc80 efficiently (*Figure 1F* and *Figure 1—figure supplement 3B*), indicating that a single negative charge at positions Thr358 and Thr360 cannot functionally replace phosphate groups in our reconstituted system. This contradicts previous results obtained with a SKA3 fragment and a GST-NUF2:NDC80 sub-complex (*Zhang et al., 2017*). Consistent with multiple phosphosites in SKA3[C] and the importance of phosphorylation for Ska:Ndc80 binding, Ska lacking SKA3[C] did not bind Ndc80 (*Figure 1F*, orange traces). Collectively, these results demonstrate, for the first time, a direct interaction between full length Ndc80 and Ska complexes, and show that phosphorylation of Thr358 and Thr360 in SKA3[C] by CDK1:Cyclin B is necessary for its formation.

## Ska binds the NDC80:NUF2 coiled coil and the Ndc80-loop is dispensable

We next set out to identify the Ndc80 regions that mediate the interaction with Ska. Phosphorylated Ska did not bind to an SPC24:SPC25 dimer, or to two engineered constructs, Ndc80[dwarf] and Ndc80[bonsai], that lack large fragments of the coiled-coils in the Ndc80 subunits (*Figure 2A*, orange traces; *Figure 2—figure supplement 1A–B*). The latter observation is at odds with a previous report that identified an interaction between Ska and Ndc80[bonsai] (*Janczyk et al., 2017*). In this previous study, Ska (without phosphorylation) and Ndc80[bonsai] had been incubated on microtubules, a condition that might expose residual, low binding affinity between these constructs. Collectively, our observations suggest a potential requirement of the NDC80:NUF2 coiled coil in Ska binding (*Figure 2B*). To test this idea, we generated Ndc80[jubaea], an extended Ndc80[bonsai] analogue that is also amenable to bacterial expression. Ndc80[bonsai] contains a total of 17% of the predicted coiled coil in all Ndc80 subunits, while Ndc80[jubaea] covers 66% of it. Importantly, Ndc80[jubaea] bound phosphorylated Ska (*Figure 2A*, blue traces, and *Figure 2—figure supplement 1C*). Collectively, these results demonstrate that NDC80[286-504]:NUF2[169-351] encompasses the Ska-binding site, and that the Ndc80 tetramerization domain is not required for the interaction with Ska.

Previously analyses identified the Ndc80-loop (NDC80[427-464], *Figure 2B–C*) as a prime candidate for Ska binding, because loop deletions or sequence inversions prevent kinetochore recruitment of Ska in vivo (*Zhang et al., 2012*; *Zhang et al., 2017*). To address the function of the Ndc80-loop directly, we designed, and successfully expressed and purified, Ndc80 constructs with a partially or entirely deleted loop region. To our surprise, these 'loopless' truncation constructs retained the ability to form a complex with Ska (*Figure 2D*). This crucial observation indicates that impaired recruitment of Ska to kinetochores in cells expressing deleted or modified Ndc80-loop sequences does not reflect impairments of the Ska-binding site, but rather a regulatory role of the Ndc80-loop that enables Ska recruitment.

## Structural characterization of the Ska:Ndc80 interaction

Both Ska and Ndc80 are highly elongated and contain flexible or disordered fragments, a challenge for high-resolution structural characterisation. Electron microscopy after low-angle metal shadowing visualized the characteristic,~80 nm long, 8-subunit Ndc80:Mis12 subcomplex of the KMN network (*Figure 2E*). However, while Ska was visible with this technique despite its small size, no conspicuous or characterizing structural features were revealed (*Figure 2E*). Conjugation of a globular tetramer incorporating one Traptavidin (T) (*Chivers et al., 2010*) and three Streptavidin subunits (abbreviated as $T_1S_3$; 88 kDa) to C-terminally biotinylated SKA1 (30 kDa) facilitated the recognition of rotary-shadowed SKA1 and revealed Ska dimers with a ~ 10 nm separation between SKA1[C] and the N-terminal coiled coils of SKA1, SKA2, and SKA3 that form the Ska dimerization interface (*Jeyaprakash et al., 2012*) (*Figure 2E* and *Figure 2—figure supplement 2A–B*). The $T_1S_3$ labeling enabled us to localize Ska bound to Ndc80:Mis12. A dimer of Ska bound a single Ndc80:Mis12 without apparently inducing multimerization of Ndc80. The position of SKA1[T1S3] near the middle of Ndc80 is consistent with

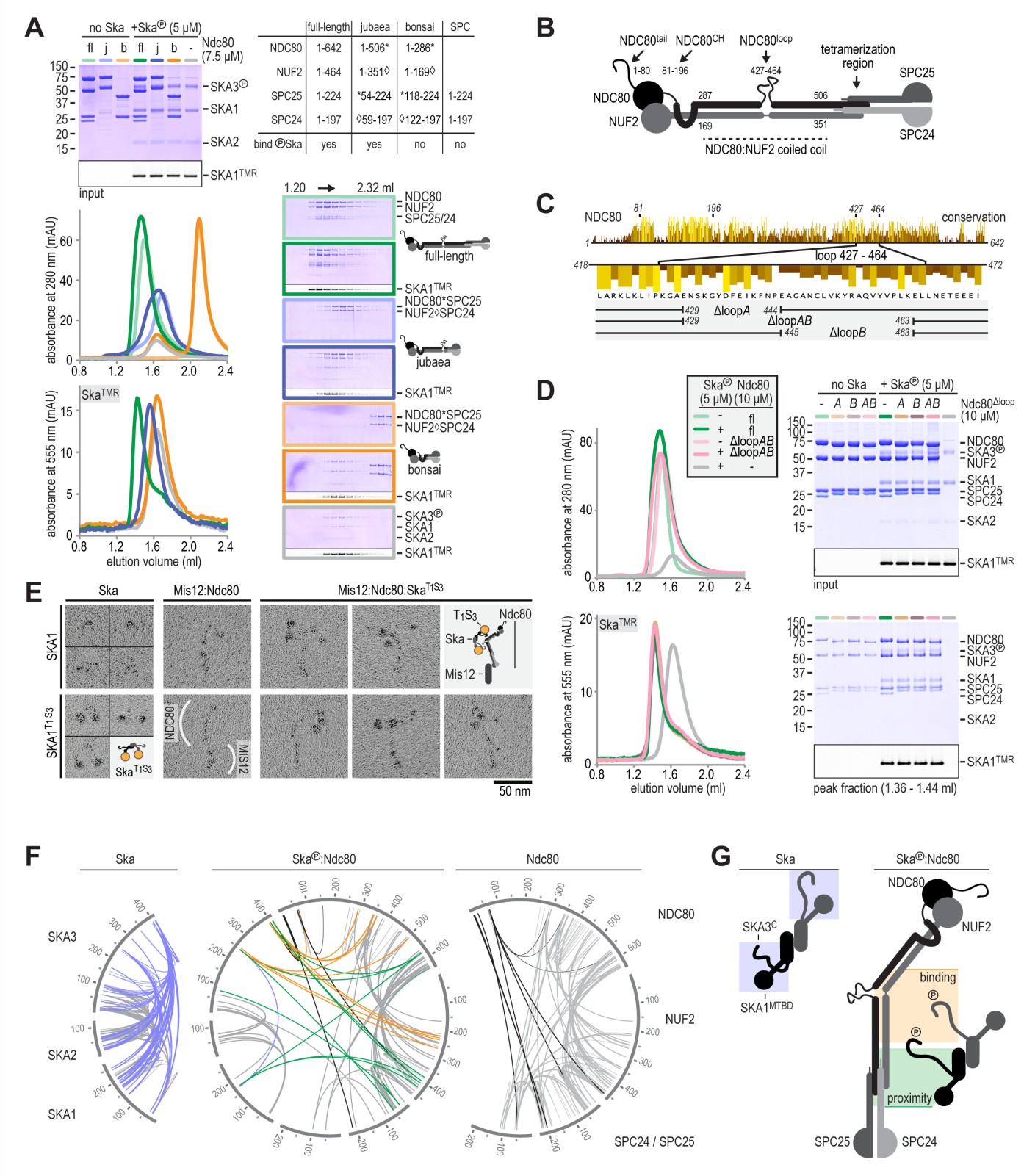

**Figure 2.** Ska binds the NDC80:NUF2 coiled-coil and the Ndc80-loop is dispensable. (**A**) Full-length (fl), jubaea (j), and bonsai (b) Ndc80 complexes were tested for their ability to bind phosphorylated Ska. Symbols (* and ◊) indicate fusion proteins in jubaea and bonsai Ndc80. (**B**) Overview of Ndc80 as in *Figure 1B*. NDC80^287-504 and NUF2^169-351, the Ska binding region that is present in Ndc80C^sequoia and absent in Ndc80C^bonsai, is indicated. (**C**) Overview of the three tested constructs that lack different parts of the NDC80-loop. The conservation of Ndc80 and its loop-region are shown. (**D**)

*Figure 2 continued on next page*

*Figure 2 continued*

Ndc80 lacking the Ndc80-loop still binds phosphorylated Ska. (E). Ska and Mis12:Ndc80:Ska were visualized by electron microscopy after glycerol-spraying and low-angle metal shadowing. SKA1$^{MTBD-biotin}$ (30 kDa) was conjugated with the biotin-binding globular T$_1$S$_3$ (88 kDa) to facilitate the recognition of Ska in micrographs. The presence of Mis12 (20 nm) marks the SPC24:SPC25 side of Ndc80 (62 nm). See *Figure 2—figure supplement 2* for detailed sample preparation information. (F) Intra- and intermolecular crosslinks for Ska, Ska:Ndc80, and Ndc80. Contacts between SKA3$^C$ and the rest of Ska are highlighted in blue. Contacts between SKA3$^C$ and the NDC80:NUF2 coiled coil and the Ndc80 tetramerization domainare shown in orange and green, respectively. The SKA1$^{MTBD}$ is also proximal to the tetramerization domain. (G) A schematic representation of proximities.

The online version of this article includes the following figure supplement(s) for figure 2:

**Figure supplement 1.** Testing Ska binding interactions with various Ndc80 constructs.
**Figure supplement 2.** Preparation of Traptavidin-Streptavidin-labelled Ska for rotary shadowing electron microscopy.
**Figure supplement 3.** Protocol of Ska:Ndc80 complex preparation for chemical cross-linking.
**Figure supplement 4.** Deletion of the MTBD of SKA1 does not interfere with the phosphorylayion-dependent binding of Ska to Ndc80.

the binding of SKA3 to the NDC80:NUF2 coiled coil and highlights how the microtubule-binding domains of the Ska dimer are positioned relative to the CH-domains of NDC80:NUF2. (*Figure 2E*).

To complement these low-resolution micrographs with a proximity map, we determined potential contacts within the Ska:Ndc80 complex using DSBU (disuccinimidyl dibutyric urea) crosslinking followed by mass spectrometry (*Pan et al., 2018*). The three datasets (Ska, Ska:Ndc80, and Ndc80) contain a total of 233 unique intramolecular and 253 unique intermolecular crosslinks (*Figure 2F*, *Figure 2—figure supplement 3*, and *Supplementary file 1c-1d*). Despite the inability to distinguish the two copies of each subunit in the Ska:Ska dimer, we can draw several conclusions from the proximity maps. First, the extensive contacts of the unstructured SKA3$^{102-412}$ with the rest of Ska largely disappear upon phosphorylation by CDK1:Cyclin B and binding to Ndc80 (*Figure 2F*, blue crosslinks). In the Ndc80-bound form, the phosphorylated SKA3$^C$ contacts the NDC80:NUF2 coiled coil and appears to reach into the portion of this coiled-coil that forms the tetramerization domain (SKA3 residues 247, 254, 394, 399, 408, 410; *Figure 2F*, orange and green crosslinks). Second, the SKA1 microtubule-binding domain (MTBD) and SKA3$^{79}$ from at least one of the Ska protomers are proximal to the Ndc80 tetramerization domain (*Figure 2F*, green crosslinks). Deletion of the MTBD of SKA1 does not interfere with Ska:Ndc80 binding (*Figure 2—figure supplement 4*), and these contacts do not reflect an essential interaction between the SKA1$^{MTBD}$ and Ndc80. Third, crosslinks between the unstructured Ndc80-tail with various regions of Ndc80 and with SKA3$^{399, 410}$ emphasize the flexibility of the entire complex and the accessibility of the Ndc80-tail (*Figure 2F*, black crosslinks). Taken together, this structural analysis combining low-angle rotary shadowing and cross-linking/mass spectrometry demonstrates that the NDC80:NUF2 coiled-coil harbours a direct binding site for SKA3 that is phosphorylated at Thr358 and Thr360, that the Ndc80-loop is dispensable for Ska recruitment in vitro, and that at least one MTBD of SKA1 in a Ska dimer is positioned near the Ndc80 tetramerization domain (*Figure 2G*).

## Aurora B does not disrupt Ska:Ndc80 binding in vitro

In previous studies, Aurora B kinase activity has been shown to counteract the recruitment of Ska to kinetochores. This crucial observation appears to link the establishment of robust microtubule attachment with the suppression of Aurora B and the recruitment of Ska (*Chan et al., 2012*; *Janczyk et al., 2017*; *Sivakumar and Gorbsky, 2017*). Aurora B kinase phosphorylates the N-terminal tail of Ndc80 and this weakens microtubule attachments (see Introduction). While previous studies advocated a requirement of the Ndc80-tail for kinetochore recruitment of Ska in vivo (*Cheerambathur et al., 2017*; *Janczyk et al., 2017*), deletion of the unstructured Ndc80-tail does not perturb binding of phosphorylated Ska to Ndc80 in our reconstituted system (*Figure 3A*). This suggests that the Ndc80-tail, like the Ndc80-loop, contributes indirectly to the recruitment of Ska by establishing a proper kinetochore-microtubule interface rather than by providing a docking site.

Aurora B also phosphorylates SKA1 and SKA3 on at least seven consensus and non-consensus sites, including four in the MTBD, but Aurora B phosphorylation has only small effects on the interaction of Ska with microtubules in vitro (*Abad et al., 2014*; *Chan et al., 2012*; *Schmidt et al., 2012*). Furthermore, enhanced Ska recruitment to kinetochores upon Aurora B inhibition is also observed when microtubules are depolymerized (*Chan et al., 2012*; *Janczyk et al., 2017*; *Sivakumar and*

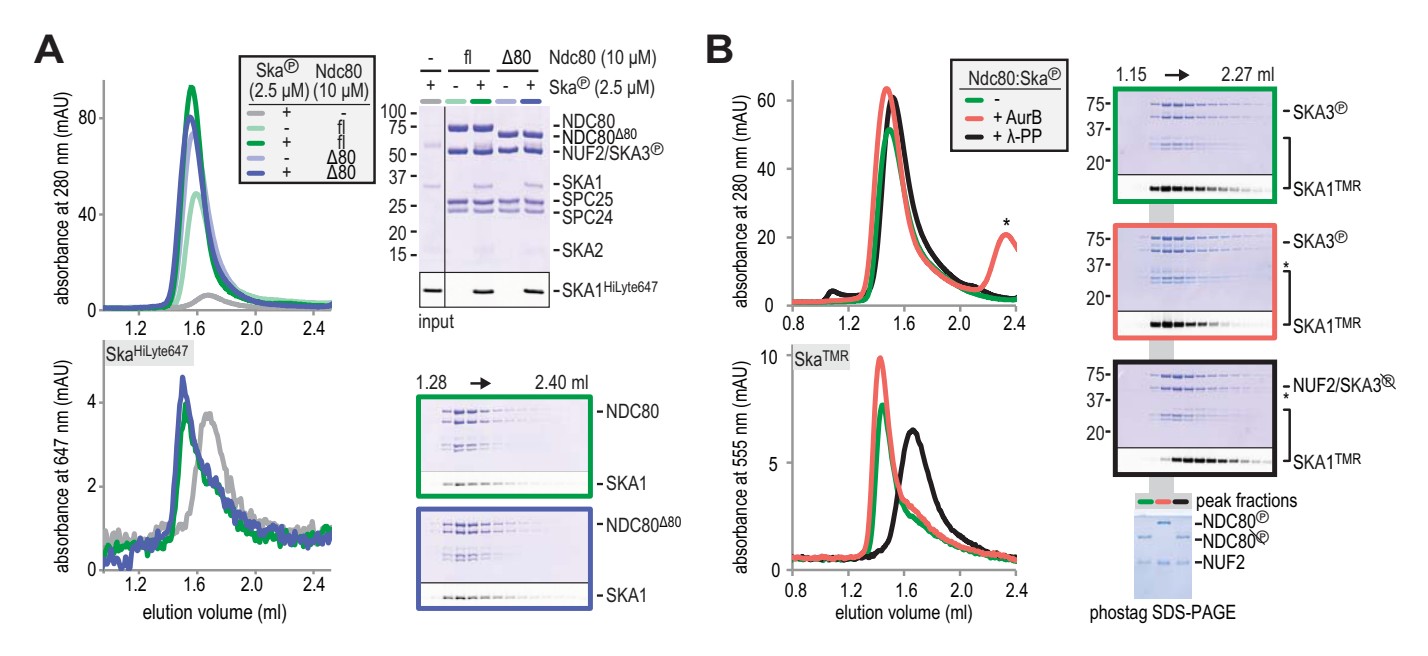

**Figure 3.** Aurora B kinase activity does not disrupt Ska:Ndc80 binding. (**A**) Full-length (fl) and tailless (Δ80) Ndc80 both bind phosphorylated Ska, as analyzed by SEC and SDS-PAGE. (**B**) A Ska:Ndc80 complex was exposed to Aurora B kinase or lambda-phosphatase. Low amounts of Ska:Ndc80 impair the detection of Ska by Coomassie but fluorescent SKA1 was detected. Dephosphorylation disrupted the Ska:Ndc80 complex but Aurora B phosphorylation did not. Kinase activity is indicated by the shift in elution volume of Aurora B treated Ndc80:Ska and by the altered migration of phosphorylated NDC80 after gel filtration on the phostag SDS-PAGE. The asterisk marks Aurora B kinase.

*Gorbsky, 2017*), arguing that Aurora B does not control Ska recruitment to kinetochores by modulating its binding affinity for microtubules.

We therefore asked if Aurora B affects the stability of the Ska:Ndc80 interaction. Exposure of pre-formed Ska:Ndc80 complex to Aurora B did not cause its dissociation. Aurora B activity was confirmed by a shift in the elution volumes of Ndc80 and Ndc80:Ska from a SEC column and by the altered migration of the NDC80 subunit in phostag SDS-PAGE (*Kinoshita et al., 2009*) (*Figure 3B*, black and red traces). Conversely, dephosphorylation of pre-formed Ska:Ndc80 complex by lambda-phosphatase displaced Ska from Ndc80. Thus, deletions of the Ndc80-tail or of the Ndc80-loop, and Aurora B activity, all of which prevent kinetochore recruitment of Ska in vivo, do not affect Ska: Ndc80 binding in vitro. This suggests that Ska recruitment is licensed by particular features of the kinetochore-microtubule interface that signal successful bi-orientation.

## The Ndc80-tail is required for end-on Ndc80-microtubule attachment

A single microtubule-binding site in the kinetochore contains multiple closely spaced Ndc80 complexes, with recent estimates converging on 6 to 8 complexes per attachment site (*Huis in 't Veld et al., 2016*; *Suzuki et al., 2015*; *Weir et al., 2016*). To address how physical clustering affects the microtubule binding properties and other interactions of Ndc80, we previously engineered an oligomerization module allowing controlled binding of 1, 2, 3 or 4 Ndc80 complexes (*Volkov et al., 2018*). We observed that multivalency has a dramatic effect on the residency time of Ndc80 on microtubules, increasing it by more than an order of magnitude for every Ndc80 added (*Volkov et al., 2018*). In force measurements with optical tweezers, sparse coating of beads with multivalent Ndc80 modules resulted in more efficient force-coupling than dense distributions of individual Ndc80 complexes (*Volkov et al., 2018*).

Our present work indicates that clustering of Ndc80 is not required to bind Ska in vitro, but we decided to investigate possible effects of this interaction on Ndc80 clusters, as these are likely to provide a more realistic representation of the kinetochore distribution of Ndc80. As before, we harnessed tetrameric $T_1S_3$-modules (like the one we used to increase the size of Ska for visualization in

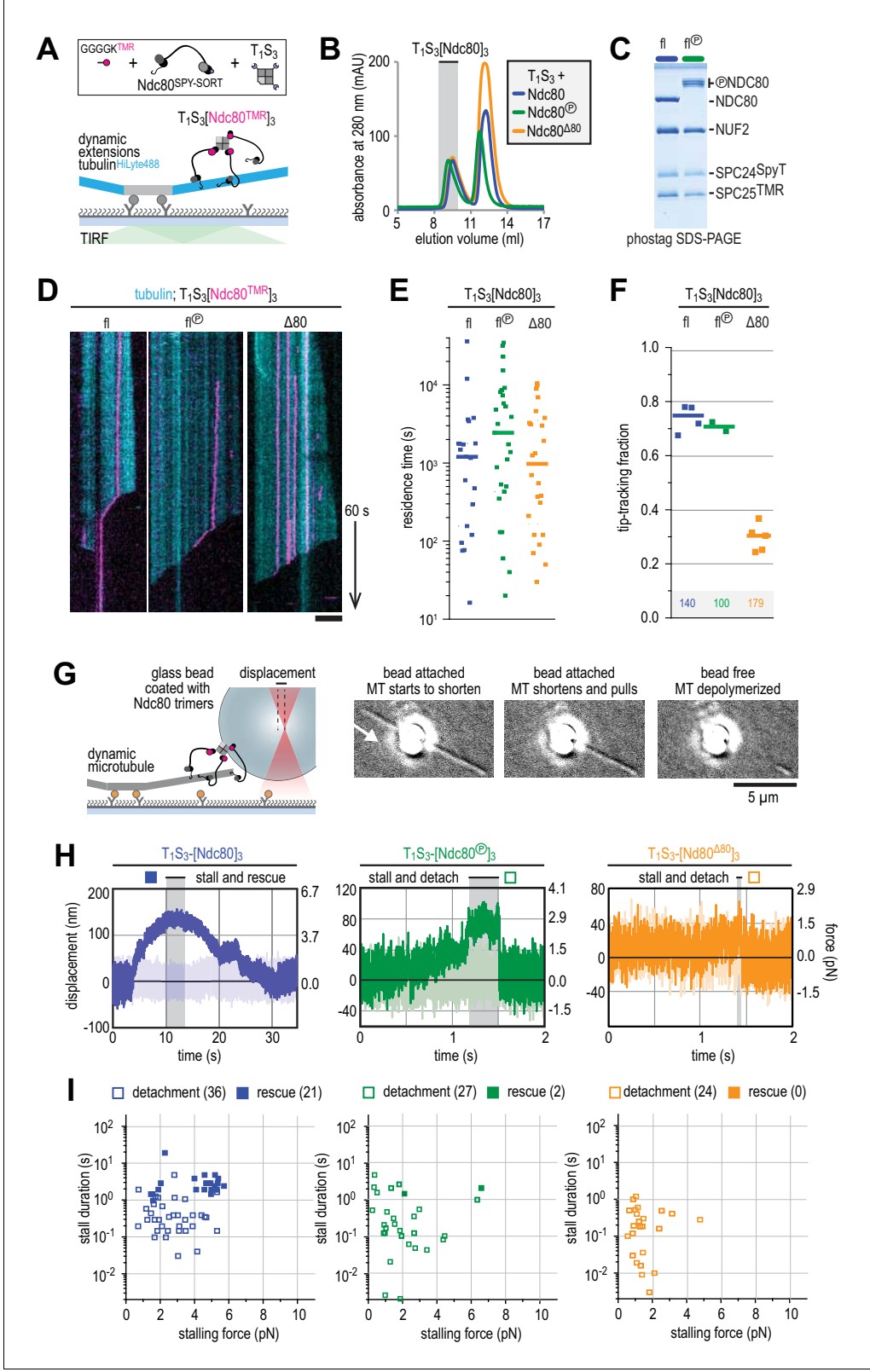

**Figure 4.** Ndc80-tail is required for interaction with a microtubule end. (**A**) Experimental setup featuring trimeric $T_1S_3[Ndc80]_3$ modules visualized on dynamic microtubules using TIRF microscopy. (**B**) Size-exclusion chromatography traces showing separation of monomeric Ndc80 from trivalent $T_1S_3$-Ndc80 modules using full-length Ndc80 (fl, blue), Aurora B phosphorylated full-length Ndc80 (fl-phosphorylated, green), and Δ80-Ndc80 (orange). (**C**) Phostag gel demonstrating successful phosphorylation of the NDC80 subunit by Aurora B. (**D**) Kymographs showing Ndc80 trimers (magenta) on

*Figure 4 continued on next page*

*Figure 4 continued*

dynamic microtubules (cyan). Scale bars: vertical (60 s), horizontal (5 μm). (**E**) Residence times of Ndc80 trimers on taxol-stabilized microtubules. Horizontal bars: median. T-tests indicate the following two-tailed p-values: $T_1S_3[Ndc80]_3$ vs $T_1S_3[Ndc80^{®}]_3$: p=0.26; $T_1S_3[Ndc80]_3$ vs $T_1S_3[Ndc80^{Δ80}]_3$: p=0.07; $T_1S_3[Ndc80^{®}]_3$ vs $T_1S_3[Ndc80^{Δ80}]_3$: p=0.63. (**F**) Fraction of Ndc80 trimers that initiate movement in the direction of microtubule shortening upon encounter with a depolymerizing end. Squares: fractions in an individual experiment, horizontal bars: median. T-tests indicate the following two-tailed p-values: $T_1S_3[Ndc80]_3$ vs $T_1S_3[Ndc80^{®}]_3$: p=0.48; $T_1S_3[Ndc80]_3$ vs $T_1S_3[Ndc80^{Δ80}]_3$: $p<10^{-5}$; $T_1S_3[Ndc80^{®}]_3$ vs $T_1S_3[Ndc80^{Δ80}]_3$: $p<10^{-3}$. (**G**) Left: setup of an optical trap experiment. Right: DIC images showing a microtubule depolymerizing past an optically trapped bead. Scale bar: 5 μm. (**H**) Representative traces of a microtubule pulling on a bead coated with Ndc80 trimers (blue), Aurora B phosphorylated wild type Ndc80 trimers (green) or trimers containing $Ndc80^{Δ80}$ (orange). (**I**) Correlation between the stalling force and the duration of the stall for each individual stall event resulting in a detachment (open symbols) or rescue (filled symbols) for Ndc80 trimers (blue), Aurora B phosphorylated Ndc80 trimers (green) or Ndc80 trimers containing $NDC80^{Δ80}$ (orange). Two-sided Fisher exact testing indicates different detachment-rescue distributions between untreated Ndc80 (36-21) and Aurora B phosphorylated (27–2, p=0.004) or $Ndc80^{Δ80}$(24–0, p=0.0002). The distribution does not differ significantly between Aurora B phosphorylated and $Ndc80^{Δ80}$ (p=0.49).

rotary shadowing experiments) to immobilize three Ndc80 complexes onto the same particle (*Figure 4A*). $T_1S_3$-$[Ndc80]_3$ modules contain three Streptavidin subunits that are covalently modified with fluorescent Ndc80, while the available Traptavidin remains available for immobilization, if needed.

We started by asking how Aurora B affects the plus end binding properties of Ndc80 in the presence of Ska. Phosphorylation of the Ndc80-tail tunes the affinity of Ndc80 for microtubules in vitro and in vivo (see for instance *Long et al., 2017*; *Zaytsev et al., 2015*). From a mechanistic perspective, however, if and how phosphorylation of the Ndc80-tail influences force-coupling with dynamic microtubules is poorly understood. To address this, we exposed trivalent Ndc80 modules to Aurora B kinase activity. Efficient phosphorylation of the Ndc80-tail was confirmed by a shift in SEC, mass spectrometry, and phostag SDS-PAGE analysis (*Figure 4B–C* and *Supplementary file 1e-1f*). We further assembled $T_1S_3$-$[Ndc80]_3$ modules containing $NDC80^{Δ80}$, that is lacking the tail altogether. Consistent with our previous observations (*Volkov et al., 2018*), trivalent Ndc80 assemblies bound to the microtubule lattice for minutes and efficiently followed depolymerizing microtubule ends (*Figure 4D*). Remarkably, Aurora B phosphorylation or truncation of the NDC80-tail did not influence the residence time of $T_1S_3$-$[Ndc80]_3$ on the microtubule lattice (*Figure 4D–E*). Conversely, truncation of the Ndc80-tail, but not its phosphorylation by Aurora B, prevented trivalent Ndc80 from following the shortening ends of microtubules (*Figure 4E–F*). Thus, these results identify the Ndc80-tail as being crucial to attach Ndc80 to depolymerizing microtubule ends. We conclude that trivalent Ndc80 modules bind the microtubule lattice stably through their CH-domains, but rely on the tails, regardless of their state of phosphorylation, to remain attached to a shortening microtubule end in the absence of a resisting force.

Previously, we used optical tweezers to study the ability of reconstituted kinetochore particles immobilized on beads to capture force generated by a depolymerizing microtubule. We found that multivalent Ndc80 modules stall microtubule depolymerization under microtubule-generated forces up to 5–6 pN (*Volkov et al., 2018*). These stalling events either induced a rescue of microtubule growth or were followed by an Ndc80-microtubule detachment event and continued microtubule depolymerization (*Volkov et al., 2018*). Using a similar experimental setup, we compared $T_1S_3$-$[Ndc80]_3$ modules with unmodified, phosphorylated, or truncated Ndc80-tails (*Figure 4G*). Consistent with the inability to tip-track shortening microtubules (*Figure 4F*), tailless trivalent Ndc80 modules ($NDC80^{Δ80}$) dissociated rapidly from depolymerizing microtubules and never rescued microtubule shortening (*Figure 4H–I*, orange trace). Force-induced stalls by unphosphorylated Ndc80 modules were followed by microtubule regrowth in 21 events, and by detachment in 36 events (*Figure 4H–I*, blue trace). In contrast, Ndc80 modules exposing a phosphorylated tail detached from microtubule ends without rescue in 27 of 29 events (*Figure 4I*, green trace). Thus, Ndc80 modules phosphorylated by Aurora B detached from shortening microtubule ends under force despite their ability to tip-track depolymerizing microtubules without load (*Figure 4F*). Unphosphorylated Ndc80 modules behaved differently and remained bound to microtubule ends independently of the force applied (*Figure 4I*, blue symbols). In line with our previous analyses (*Volkov et al., 2018*), there was a correlation between the force at stall and the likelihood of a rescue (*Figure 4I*). In addition, in this and subsequent experiments described in *Figure 5*, the duration

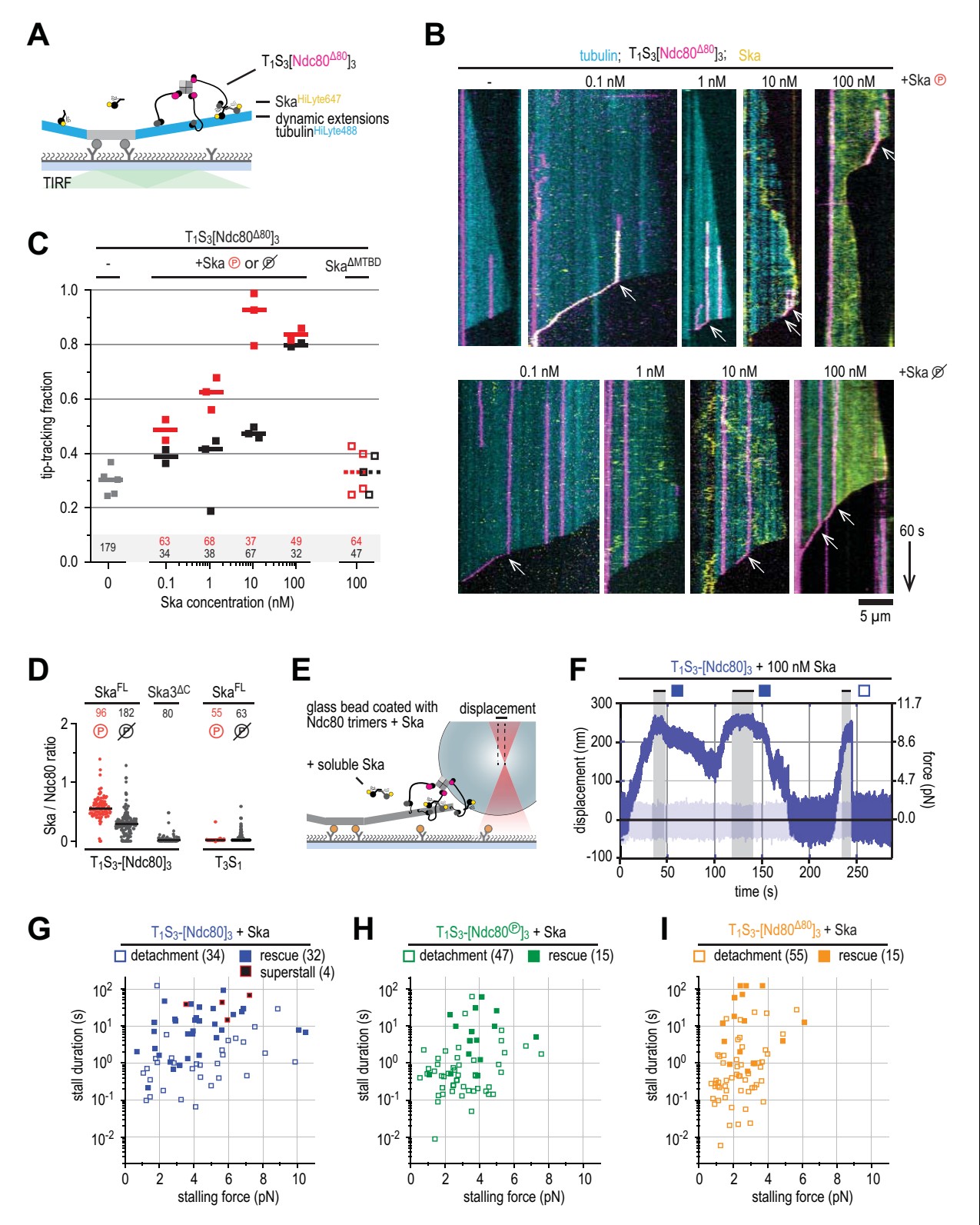

**Figure 5.** Ska bound to Ndc80 presents an additional microtubule end-binding site that stabilizes the stalled microtubule ends. (A) Ska and trimeric Ndc80$^{\Delta 80}$ modules were imaged simultaneously on dynamic microtubules using TIRF microscopy. (B) Trimeric Ndc80$^{\Delta 80}$ modules (magenta) on dynamic microtubules (cyan) in the presence of increasing concentrations of phosphorylated (top row) or dephophosphorylated (bottom row) Ska (yellow). Scale bars: vertical (60 s), horizontal (5 µm). Arrows indicate successful end-tracking events. (C) Fraction of trimeric Ndc80$^{\Delta 80}$ modules that initiate movement

*Figure 5 continued on next page*

*Figure 5 continued*

in the direction of microtubule shortening upon encounter with a depolymerizing end in the presence of Ska. Squares: fractions in an individual experiment, horizontal bars: median. (D) Ratio of Ska to Ndc80 after incubation of the beads coated with Ndc80 trimers and then Ska (400 nM; see also *Figure 5—figure supplement 1*). Horizontal lines: median. A t-test indicates significant different between the addition of phosphorylated or dephosphorylated Ska: $T_1S_3[Ndc80]_3$ + Ska vs Ska℗: $p<10^{-17}$. Other two-tailed p-values are $T_1S_3[Ndc80]_3$ + Ska vs Ska3$^{\Delta C}$: $p<10^{-22}$; $T_1S_3[Ndc80]_3$ + Ska℗ vs Ska3$^{\Delta C}$: $p<10^{-34}$; $T_3S_1$ + Ska vs Ska℗: $p=0.012$; $T_3S_1$ + Ska vs Ska3$^{\Delta C}$: $p=0.99$; $T_3S_1$ + Ska℗ vs Ska3$^{\Delta C}$: $p=0.012$; Ska℗ + $T_1S_3[Ndc80]_3$ vs $T_3S_1$: $p<10^{-28}$; Ska + $T_1S_3[Ndc80]_3$ vs $T_3S_1$: $p<10^{-30}$. (E) Optically trapped bead coated with Ndc80 trimers and Ska in a chamber with dynamic microtubules and additional soluble Ska. (F) An example force trace obtained in the presence of dephosphorylated Ska and a bead coated with non-phosphorylated Ndc80 trimers. (G–I) Correlation between the stalling force and the duration of the stall for each individual stall event in the presence of 10–100 nM Ska resulting in a detachment (open symbols), rescue (filled symbols) or superstall (black symbols) for the beads coated with non-phosphorylated Ndc80 trimers (G, blue symbols), Aurora B phosphorylated Ndc80 trimers (H), green symbols) or trimers containing Ndc80$^{\Delta 80}$ (i), orange symbols). Two-sided Fisher exact testing indicates different detachment-rescue distributions between untreated Ndc80 (34-32) and Aurora B phosphorylated (47–15, p=0.006) or Ndc80$^{\Delta 80}$(55–15, p=0.001). The addition of Ska (in comparison to *Figure 4I*) did change the detachment-rescue distribution for Ndc80$^{\Delta 80}$ (p=0.010), but not significantly for untreated and Aurora B phosphorylated Ndc80 (p=0.207 and 0.081, respectively). See *Figure 5—figure supplements 2* and *3* for data separated per Ska concentration and phosphorylation state.

The online version of this article includes the following figure supplement(s) for figure 5:

**Figure supplement 1.** Ratio of Ska copy number to Ndc80 copy number after incubation of beads coated with untreated, Aurora-B-phosphorylated or tail-less Ndc80 trimers, and then Ska in indicated concentration.

**Figure supplement 2.** Correlations between the stalling force and the duration of the stall for each individual stall event in the presence of 10 or 100 nM Ska resulting in a detachment (open symbols), rescue (filled symbols) or superstall (black symbols) for the beads coated with non-phosphorylated Ndc80 trimers (A), Aurora B-phosphorylated Ndc80 trimers (B), or tail-less Ndc80 trimers (C).

**Figure supplement 3.** Distributions of durations and forces of stalls with Ndc80 and plus or minus Ska.

of the stall emerged as an apparently critical parameter in determining the likelihood of a rescue after stall. Specifically, we did not observe rescues for stalls that ended within ~1 s, even for high stall forces. Rescues were only observed for longer stalls, albeit not as an obligate outcome, because detachments were also observed (*Figure 4I*).

## Ska stabilizes end-on Ndc80-microtubule interactions under force

Next, we asked if and how Ska influences the interaction of Ndc80 with microtubule ends. Since trivalent Ndc80 modules are very efficient microtubule tip-trackers by themselves in our assays (*Figure 4F*), we added fluorescently labeled Ska to flow chambers with dynamic microtubules and trivalent Ndc80$^{\Delta 80}$ modules, which are instead very poor end-trackers (*Figure 5A*). CDK1-phosphorylated Ska associated with lattice-bound and tip-tracking Ndc80$^{\Delta 80}$ modules when Ska was added at concentrations as low as 100 pM (*Figure 5B–C*). At concentrations of 1 and 10 nM, phosphorylated Ska effectively conferred tip-tracking ability to the Ndc80$^{\Delta 80}$ modules whereas dephosphorylated Ska did not. These results demonstrate that upon binding to Ndc80, Ska creates an additional microtubule binding site which enables end-tracking of Ndc80$^{\Delta 80}$:Ska complexes.

The addition of Ska at 100 nM conferred end-tracking ability to trivalent Ndc80$^{\Delta 80}$ modules whether Ska was phosphorylated or not (*Figure 5B–C*). This was not observed in the presence of Ska lacking the SKA1$^{MTBD}$ at 100 nM and the stabilizing effect of Ska did thus strictly require its ability to interact with microtubules (*Figure 5C*). The stabilization of end-on Ndc80-microtubule interactions in the presence of non-phosphorylated Ska indicates that some Ska can bind Ndc80 without the phosphorylation of SKA3 in the context of trivalent Ndc80 modules. Consistent with this explanation, we did observe binding of non-phosphorylated Ska to beads coated with trivalent Ndc80 complexes (but not to beads coated with $T_3S_1$ without Ndc80). This binding was weaker than for phosphorylated Ska but dependent on the SKA3$^C$ region: Ska complexes lacking SKA3$^C$ failed to bind Ndc80 trimers on the beads (*Figure 5D*, *Figure 5—figure supplement 1*).

We next set out to test if the presence of Ska influences force-coupling and kinetochore-microtubule attachments. For this purpose, we added Ska:Ndc80-coated glass beads to dynamic microtubules and added soluble Ska at a concentration of 10 or 100 nM (*Figure 5E*). Real-time monitoring of Ska in the DIC-based optical-tweezers setup is not feasible, and we could therefore not distinguish if Ska was present at the Ndc80-microtubule interface during force recordings. We note that Ska associates with the microtubule lattice at these concentrations (*Figure 5B*) and binds to Ndc80-coated glass beads (*Figure 5D*). Since we have not been able to observe differences between the addition of phosphorylated and non-phosphorylated Ska under these conditions (*Figure 5—figure*

*supplements 2–3*), we pooled observations from both forms and from both concentrations of Ska and set out to investigate the effects of Ska on force-coupling.

The presence of Ska resulted in remarkably long force-dependent stalls of microtubule depolymerization (a typical example is shown in *Figure 5F*). This required the presence of both the microtubule binding domain of SKA1 and the C-terminal tail of SKA3 (*Figure 5—figure supplement 3*), indicating that the stabilization of Ndc80-mediated microtubule stalls by Ska requires SKA3-Ndc80 and SKA1-microtubule interactions. Longer stalls in the presence of Ska were observed for all three different Ndc80-tail constructs, albeit at different forces and with different rescue probabilities (*Figure 5G–I*). While stalls for intact unphosphorylated Ndc80 in absence of Ska were limited to 5 s (in 56 of 57 cases, *Figure 4I*), stall durations for the intact Ndc80 in presence of Ska exceeded 5 s in 25 out of 66 cases (*Figure 5G* and *Figure 5—figure supplements 2* and *3*). We also observed a fraction of Ska:Ndc80 beads that stalled microtubules at forces reaching ~10 pN, exceeding the limit of 6 pN observed in absence of Ska (*Figure 5G*).

In some cases, the presence of 100 nM Ska resulted in force-induced stalls that persisted for tens of seconds and were deliberately ended by increasing the stiffness of the trap and detaching the Ska:Ndc80-coated bead from the microtubule end. After bead detachment, these microtubules failed to undergo rescue or disassembly, as if they obtained a hyper-stable, frozen state during the stall. This condition was apparently independent of Ska:Ndc80 interactions, because it was also observed in presence of Ska lacking the SKA3$^C$. These hyper-stable microtubules were only observed when Ska at 100 nM and an unphosphorylated Ndc80-tail were combined (7 out of 57 events; *Figure 5—figure supplement 3*). This suggests that the presence of Ska at high concentrations stabilizes microtubule ends that are stalled under force by Ndc80 (*Figure 5G*, *Figure 5—figure supplement 3*).

The presence of Ska resulted in force-induced stalls of microtubule depolymerization that frequently lasted longer than a second (*Figure 5G–I*). Importantly, Ndc80 with phosphorylated or deleted Ndc80-tails was almost never able to stall the ends of depolymerizing microtubules for these lengths in the absence of Ska (*Figure 4I*). Increased stall durations in the presence of Ska correlated with an increase in the number of microtubules undergoing rescue during force-induced stalls by Ndc80 modules with phosphorylated or deleted NDC80-tails (*Figure 5H–I*, *Figure 6*).

## Discussion

Kinetochore recruitment of Ska is a late mitotic event that signals the completion of bi-orientation and the establishment of kinetochore tension (*Auckland et al., 2017*; *Chan et al., 2012*; *Gaitanos et al., 2009*; *Hanisch et al., 2006*; *Raaijmakers et al., 2009*; *Welburn et al., 2009*; *Zhang et al., 2017*). The precise molecular requirements for kinetochore recruitment of Ska, however, had not been identified, motivating the present analysis. To shed light on this mechanism, we developed assays that, for the first time, allowed us to identify and dissect a direct interaction of the full-length versions of these complexes. We conclude that 1) CDK1-mediated phosphorylation of SKA3 promotes the formation of a stable Ska:Ndc80 complex; 2) this requires phosphorylation of Thr358 and Thr360 in SKA3; 3) the NDC80:NUF2 coiled-coils, but not the Ndc80-tail or -loop, are necessary for the Ska:Ndc80 interaction.

We observed that the requirement for SKA3 phosphorylation by CDK1 is, in the presence and absence of microtubules, attenuated in the context of Ndc80 multimerization (*Figure 5C*, *Figure 5—figure supplement 1*). This might explain why phosphomimetic substitutions in SKA3 are sufficient for loading Ska onto kinetochores in vivo (*Zhang et al., 2017*), where Ndc80 is oligomerized, compared to our SEC experiments with monomeric Ndc80 (*Figure 1F*). In line with previous studies that assessed Ndc80:Ska binding on microtubules with monomeric Ndc80 (*Helgeson et al., 2018*) or fragments thereof (*Schmidt et al., 2012*; *Chakraborty et al., 2019*), these observations suggest that microtubules facilitate an interaction of unphosphorylated Ska and Ndc80. However, microtubule binding is not a strict requirement for kinetochore localization of Ska, as the latter localizes to KTs, but not to spindle MTs, also upon deletion of the SKA1 microtubule-binding domain (*Abad et al., 2014*; *Schmidt et al., 2012*).

Our conclusions partly agree with a previous analysis showing that CDK1-mediated phosphorylation of the C-terminal region of SKA3 is important for Ska:Ndc80 binding (*Zhang et al., 2017*). Similarly to that study, we find that Ndc80$^{bonsai}$ does not bind Ska, an observation that contradicts

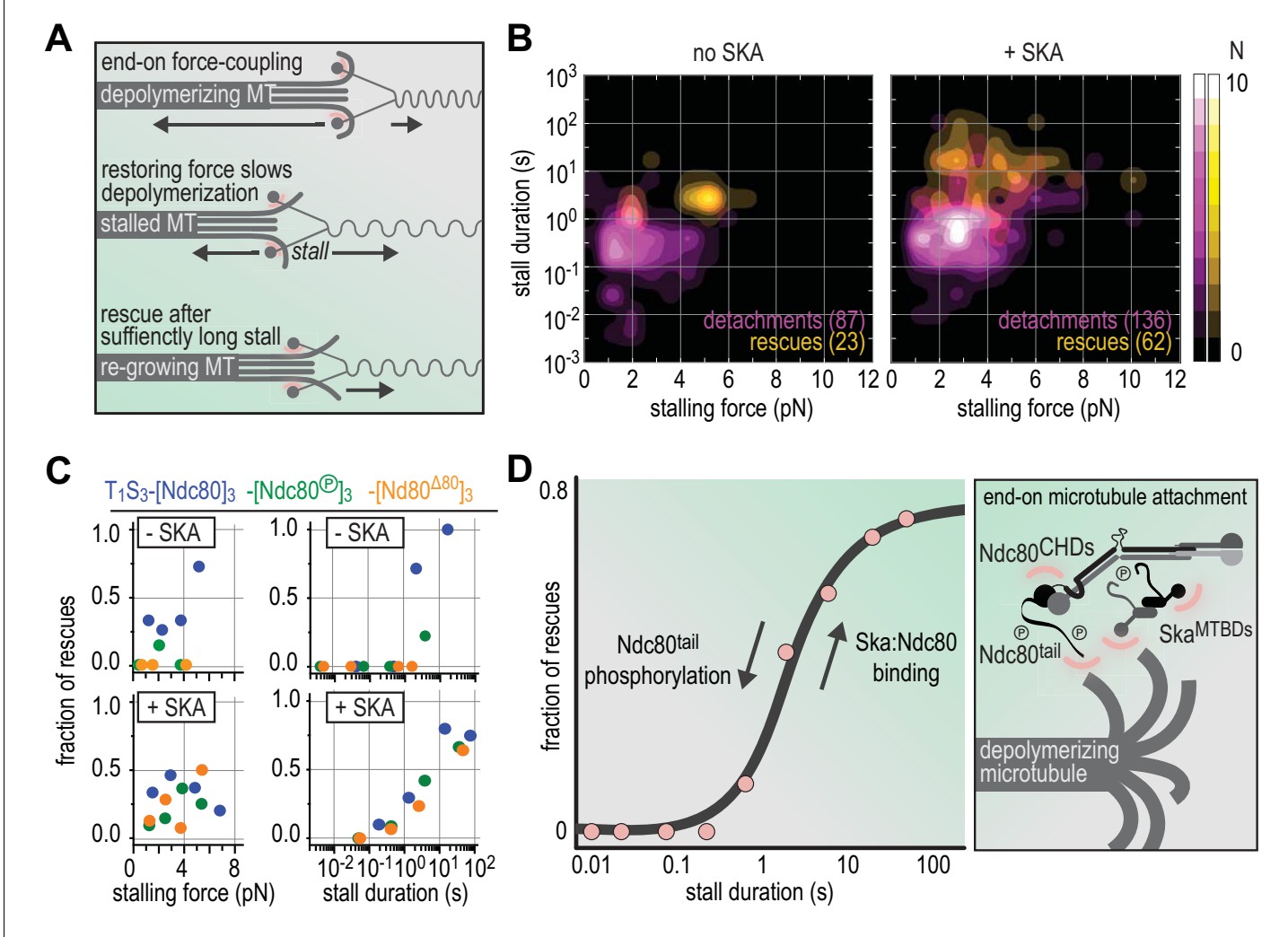

**Figure 6.** Molecular determinants of the Ska-Ndc80 interaction and their influence on microtubule tracking and force-coupling. (**A**) Schematic representation of force-coupling before, during, and after a force-induced stall of microtubule depolymerization that was followed by microtubule re-growth. (**B**) A density plot of stall durations and forces resulting in detachment or rescue in the absence of Ska (left) or in the presence of 10–100 nM Ska (right). Data are pooled for all three types of Ndc80 trimers. (**C**) The fraction of rescues was plotted against stalling force or stall duration after binning of data from the different Ndc80 complexes in the presence or absence of 10–100 nM Ska (as shown in **Figures 4** and **5**). (**D**) The fraction of rescues was plotted against stall duration after pooling and binning of all data. Detachment is more likely in the absence of Ska and when the Ndc80-tail is phosphorylated. An unphosphorylated Ndc80-tail and the presence of Ska increase the attachment survival rate.

another recent report (*Janczyk et al., 2017*). On the other hand, *Zhang et al. (2017)* concluded that Ska interacts with the Ndc80-loop, which is required for Ska recruitment in vivo (*Zhang et al., 2012*). We show here that both the Ndc80-loop and the Ndc80-tail, another Ndc80 region previously implicated in the interaction with Ska (*Cheerambathur et al., 2017*), are dispensable for the Ska:Ndc80 interaction. Instead, Ska interacts with the coiled-coils of NDC80:NUF2 (*Figure 2G*). The role of the Ndc80 coiled-coils in Ska binding agrees with a recent report in which the interaction of Ska and Ndc80 had been studied in the absence of phosphorylation but in presence of microtubules (*Helgeson et al., 2018*).

The direct and robust interaction of Ska with Ndc80 in the absence of tension suggests that tension is not required to expose an otherwise cryptic binding site for Ska on Ndc80, although we cannot exclude that the binding affinity of Ska for Ndc80 increases under force. Robust kinetochore localization of Ska in cells with depolymerized microtubules when Aurora B is inhibited also argues against a direct role of force in Ska:Ndc80 interaction (*Chan et al., 2012*; *Redli et al., 2016*;

*Sivakumar and Gorbsky, 2017*). Aurora B kinase is believed to be exquisitely sensitive to kineto-chore tension, and these observations renew the interest in the mechanisms allowing it to control Ska localization (*Krenn and Musacchio, 2015*). We show in vitro that Aurora B activity does not disrupt the Ska:Ndc80 interaction, despite successful phosphorylation of Ska and Ndc80, suggesting that Aurora B plays an indirect role on the Ska:Ndc80 interaction. It is notable that another condition shown to prevent bi-orientation, the deletion of the Ndc80-loop, also prevents Ska recruitment to kinetochores without disrupting the potential for this interaction in vitro. Collectively, these observations point to the Ndc80:Ska interaction as an effector of a separate, yet unidentified node of tension sensing in the kinetochore, and future studies will have to address the mechanistic basis of this phenomenon, including the exact role of Aurora B and of the postulated feedback mechanisms identifying Ska as an Aurora B activator (*Redli et al., 2016*), possibly through interactions with protein phosphatase 1 (PP1) (*Janczyk et al., 2017*; *Schmidt et al., 2012*; *Sivakumar and Gorbsky, 2017*; *Sivakumar et al., 2016*; *Welburn et al., 2009*).

With the demonstration that Ska and Ndc80 interact directly in a single complex, a crucial question is how this interaction affects the kinetochore-microtubule interface (*Figure 6A*). In previous studies, Ska (without phosphorylation) was shown to promote the ability of Ndc80 to track the depolymerizing ends of microtubules (*Helgeson et al., 2018*; *Schmidt et al., 2012*). The significance of this observation, however, is partly unclear, because Ndc80 can track depolymerizing ends of microtubules when part of an oligomer, its normal condition within kinetochores (*Powers et al., 2009*; *Volkov et al., 2018*). Furthermore, either complex has been shown to be able to form load-bearing attachments to microtubules in isolation (*Helgeson et al., 2018*; *Powers et al., 2009*; *Tien et al., 2010*; *Volkov et al., 2018*). When added to Ndc80, Ska was shown to increase the survival probability of Ndc80 connections with microtubules, but independently of applied force, possibly reflecting lattice stabilization at high Ska concentrations rather than the direct interaction of Ska and Ndc80 (*Helgeson et al., 2018*).

An ongoing challenge for in vitro studies is to investigate the kinetochore-microtubule interface as a unit, using systems that approximate stoichiometry, composition, and regulation of real attachment sites. Here, after identifying conditions that promote Ska:Ndc80 binding, we combined phosphorylated Ska to Ndc80 trimers that we have previously characterized as microtubule end trackers and good force couplers (*Volkov et al., 2018*). An original conclusion of our studies is that, after oligomerization of Ndc80, the NDC80 N-terminal tail is dispensable for robust binding to the microtubule lattice but is required for tracking a depolymerizing microtubule end in the absence of force. Deletion of the Ndc80-tail or its phosphorylation by Aurora B impaired the ability of Ndc80 to stall and rescue microtubule shortening and promoted detachment from microtubule ends under force.

End-on attachments in vitro can stall microtubule depolymerization in a force-dependent manner and trigger a switch to a growing state. In case of unphosphorylated Ndc80 complexes, rescue events were more likely for high stalling forces, as reported previously (*Volkov et al., 2018*). Here, we discovered that such rescue events only occurred after stall duration reached a threshold of approximately ~1 s (*Figure 4I*, *Figure 6B*). This temporal threshold appeared to persist in the presence of Ska, whereas forces required for producing rescue events in the presence of Ska varied widely (*Figure 6C*). Binning of data obtained with the entire range of tested Ndc80 and Ska complexes (*Supplementary file 1g*) readily revealed stall duration as a good indicator for the binary outcome of a force-induced stall (detachment or rescue) (*Figure 6D*). We therefore speculate that the threshold of ~1 s reflects a general property of a force-induced switch from a depolymerization to polymerization, possibly the time needed to stabilize protofilaments mechanically and assemble a growth-supportive GTP-tubulin cap. The presence of Ska increased the overall duration of microtubule stalls, possibly through stabilization of microtubule plus-ends in the stalled state. We speculate that, in our experimental setup and in vivo, Ska stabilizes kinetochore-proximal protofilaments in a force- or curvature-dependent manner. Although many molecular details remain unclear, the idea that Ska both requires and promotes force-coupling at the kinetochore-microtubule interface is consistent with gradual and tension-dependent recruitment of Ska to kinetochores during chromosome congression (*Auckland et al., 2017*). While phosphorylation of the SKA3 C-terminal region was required for Ndc80 binding in solution (*Figure 1D*) and stimulated binding on beads (*Figure 5D*), our force measurements with these beads in optical tweezers did not identify obvious effects of CDK1-phosphorylation of Ska (*Figure 5G–I*). While we do not have a clear explanation for this, we surmise that it may reflect the specific configuration of Ndc80 binding sites in our reconstituted

system. The precise spatial and temporal regulation of Ska recruitment to kinetochores in vivo and its relationship to the establishment of tension is a topic for further investigation.

Kinetochore-microtubule interactions need to be reversible in the absence of tension and stabilized upon bi-orientation. Although the reconstitution of a *bona fide* tension-sensitive kinetochore-microtubule interface requires additional components and remains a long-term goal, our data in the absence of Ska recapitulate tension-stabilized kinetochore-microtubule attachments. These results establish the N-terminal tail of Ndc80 as a crucial force-coupling element, demonstrate that phosphorylation of the Ndc80-tail by Aurora B ensures reversible and tension-sensitive kinetochore-microtubule interactions, and provide mechanistic insight into the well-described in vivo effects of mutations that mimic constitutively phosphorylated or unphosphorylated Ndc80-tails. How phosphorylation of the Ndc80-tail and Ska levels at the kinetochore are tuned in a tension-sensitive manner and whether phosphatases play a role remain open questions of great interest.

# Materials and methods

## Key resources table

| Reagent type (species) or resource | Designation | Source or reference | Identifiers | Additional information |
|---|---|---|---|---|
| Recombinant DNA reagent | pBIG1 Ska | This study | | Following the biGBac system (**Weissmann et al., 2016**): $SKA1^{SORT-HIS}$ (CasI), SKA2 (CasII), SKA3 (CasII) |
| Recombinant DNA reagent | pBIG1 $Ska^{\Delta MTBD}$ | This study | | $SKA1^{1-108\ SORT-HIS}$ (CasI), SKA2 (CasII), SKA3 (CasII) |
| Recombinant DNA reagent | pBIG1 $Ska^{3\Delta C}$ | This study | | $SKA1^{SORT-HIS}$ (CasI), SKA2 (CasII), $SKA3^{1-101}$ (CasII) |
| Recombinant DNA reagent | pBIG1 $Ska^{3\Delta C\Delta MTBD}$ | This study | | $SKA1^{1-108\ SORT-HIS}$ (CasI), SKA2 (CasII), $SKA3^{1-101}$ (CasII) |
| Recombinant DNA reagent | pBIG1 $Ska^{3\Delta C}$ | This study | | $SKA1^{SORT-HIS}$ (CasI), SKA2 (CasII), $SKA3^{1-101}$ (CasII) |
| Recombinant DNA reagent | pBIG1 $Ska^{T358AT360A}$ | This study | | $SKA1^{SORT-HIS}$ (CasI), SKA2 (CasII), $SKA3^{T358AT360A}$ (CasII) |
| Recombinant DNA reagent | pBIG1 $Ska^{T358DT360D}$ | This study | | $SKA1^{SORT-HIS}$ (CasI), SKA2 (CasII), $SKA3^{T358DT360D}$ (CasII) |
| Recombinant DNA reagent | pBIG1 Ndc80 | Musacchio laboratory, **Huis in 't Veld et al., 2016** | | NDC80 (CasI), NUF2 (CasII), $SPC25^{HIS}$ (CasIII), SPC24 (CasIV) |
| Recombinant DNA reagent | pBIG1 Ndc80 | Musacchio laboratory, **Volkov et al., 2018** | | NDC80 (CasI), NUF2 (CasII), $SPC25^{SORT-HIS}$ (CasIII), $SPC24^{SPY}$ (CasIV) |
| Recombinant DNA reagent | pBIG1 Ndc80 | Musacchio laboratory, **Volkov et al., 2018** | | NDC80 (CasI), NUF2 (CasII), $SPC25^{SORT-HIS}$ (CasIII), $SPC24^{SPY}$ (CasIV) |
| Recombinant DNA reagent | pBIG1 $Ndc80^{\Delta 80}$ | this study | | $NDC80^{\Delta 1-80}$ (CasI), NUF2 (CasII), $SPC25^{SORT-HIS}$ (CasIII), $SPC24^{SPY}$ (CasIV) |
| Recombinant DNA reagent | pBIG1 $Ndc80^{\Delta loopA}$ | this study | | $NDC80^{\Delta 429-444}$ (CasI), NUF2 (CasII), $SPC25^{SORT-HIS}$ (CasIII), $SPC24^{SPY}$ (CasIV) |
| Recombinant DNA reagent | pBIG1 $Ndc80^{\Delta loopB}$ | this study | | $NDC80^{\Delta 445-463}$ (CasI), NUF2 (CasII), $SPC25^{SORT-HIS}$ (CasIII), $SPC24^{SPY}$ (CasIV) |

*Continued on next page*

Continued

| Reagent type (species) or resource | Designation | Source or reference | Identifiers | Additional information |
|---|---|---|---|---|
| Recombinant DNA reagent | pBIG1 Ndc80$^{\Delta loopAB}$ | this study | | NDC80$^{\Delta 429-463}$ (CasI), NUF2 (CasII), SPC25$^{SORT-HIS}$ (CasIII), SPC24$^{SPY}$ (CasIV) |
| Recombinant DNA reagent | pGEX 2-rbs Ndc80$^{bonsai}$ | Musacchio laboratory, *Ciferri et al., 2008* | | $^{GST-PreScission}$NUF2$^{1-169}$:SPC24$^{122-197}$(CasI), NDC80$^{1-286}$:SPC25$^{118-224}$ (CasII) |
| Recombinant DNA reagent | pGEX 2-rbs Ndc80$^{jubaea}$ | this study | | $^{GST-PreScission}$NUF2$^{1-351}$:SPC24$^{59-197}$(CasI), NDC80$^{1-506}$:SPC25$^{54-224}$ (CasII) |
| Recombinant DNA reagent | pGEX 2-rbs Ndc80$^{dwarf}$ | this study | | $^{GST-PreScission}$NUF2$^{1-169\ \&\ 395-464}$ (CasI), NDC80$^{1-286\ \&\ 551-642}$ (CasII) |
| Recombinant DNA reagent | pGEX 2-rbs SPC25:SPC24 | Musacchio laboratory, *Ciferri et al., 2005* | | $^{GST-PreScission}$SPC25 (CasI), SPC24 (CasII) |
| Recombinant DNA reagent | pET30b-7M-SrtA | Hidde Ploegh laboratory | addgene 51141 | Sortase (Ca$^{2+}$ independent) |
| Recombinant DNA reagent | pET21a-Traptavidin | Mark Howarth laboratory | addgene 26054 | Core Traptavidin (T) |
| Recombinant DNA reagent | pET21a-DCatch | Mark Howarth laboratory | addgene 59547 | Biotin-binding dead streptavidin-SpyCatcher (S) |
| Peptide, recombinant protein | Cdk1:Cyclin-B kinase | Musacchio laboratory, *Huis in 't Veld et al., 2016* | | |
| Peptide, recombinant protein | Aurora B kinase | Musacchio laboratory, *Girdler et al., 2008* | | Aurora B$^{45-344}$: INCENP$^{835-903}$ |
| Peptide, recombinant protein | Lambda-phosphatase | Generated in-house | | |
| Peptide, recombinant protein | GGGG[Lys-HiLyte647] | Life Technologies | | |
| Peptide, recombinant protein | GGGG[Lys-TMR] | GenScript | | |
| Peptide, recombinant protein | GGGG[Lys-PEG2-Biotin][Lys-TMR] | GenScript | | |

## Expression and purification of Ska

Expression cassettes from pLIB vectors containing SKA1$^{SORT-HIS}$, SKA2, and SKA3 were combined on a pBIG1 vector using Gibson assembly as described (*Weissmann et al., 2016*). See the Key Resources Table for the different constructs used. Baculoviruses were generated in Sf9 insect cells and used for protein expression in Tnao38 insect cells. Between 60 and 72 hr post-infection, cells were washed in PBS (10 mM Na$_2$HPO$_4$, 1.8 mM KH$_2$PO$_4$, 2.7 mM KCl, 137 mM NaCl, pH 7.4) and stored at −80°C. All subsequent steps were performed on ice or at 4°C. Cells were thawed and resuspended in lysis buffer (20 mM Tris-HCl, pH 8.0, 150 mM NaCl, 10% v/v glycerol, 2 mM TCEP, 20 mM imidazole, 0.5 mM PMSF, and protease-inhibitor mix HP Plus (Serva)), lysed by sonication, and cleared by centrifugation at 108,000 g for 60 min. The cleared lysate was filtered (0.8 μM) and applied to 5 or 10 ml HisTrap FF (GE Healthcare) equilibrated in washing buffer (20 mM Tris-HCl, pH 8.0, 150 mM NaCl, 10% v/v glycerol, 1 mM TCEP, 20 mM imidazole), washed with ca. 35 column volumes, and eluted with two column volumes of elution buffer (washing buffer with 400 mM imidazole). Relevant fractions were pooled, diluted 5-fold with buffer A (20 mM Tris-HCl, pH 8.0, 30 mM NaCl, 5% v/v glycerol, 1 mM TCEP) and applied to a 25 ml Source15Q (GE Healthcare) strong anion exchange column equilibrated in buffer A. Bound proteins were eluted with a linear gradient from 30 mM to 500 mM NaCl in 180 ml. Relevant fractions were concentrated in 10 kDa molecular mass cut-off Amicon concentrators (Millipore) and applied to a Superdex 200 16/60 column (GE

Healthcare) equilibrated in 20 mM Tris-HCl, pH 8.0, 150 mM NaCl, 5% v/v glycerol, 1 mM TCEP. Relevant fractions were pooled, concentrated, flash-frozen in liquid nitrogen, and stored at −80°C. Ska complexes with deletions or point mutations were purified in the same way.

## Expression and purification of Ndc80$^{dwarf}$, Ndc80$^{jubaea}$, Ndc80$^{bonsai}$, and SPC24/25

*E. coli* BL21(DE3)-Codon-plus-RIPL cells containing the Ndc80$^{dwarf}$ or Ndc80$^{jubaeae}$ pGEX-6P-2rbs vector were grown at 37°C in Terrific Broth in the presence of Chloramphenicol and Ampicillin to an OD600 of ∼0.8. Protein expression was induced by the addition of 0.4 mM IPTG and cells were incubated ∼14 hr at 18°C. Cells were washed in PBS and pellets were stored at −20°C or −80°C. All subsequent steps were performed on ice or at 4°C. Cells were thawed and resuspended in lysis buffer (50 mM Hepes, pH 8.0, 500 mM NaCl, 10% v/v glycerol, 2 mM TCEP, 1 mM EDTA, 0.5 mM PMSF, protease-inhibitor mix HP Plus (Serva)), lysed by sonication and cleared by centrifugation at 75,600 or 108,000 g for 60 min. The cleared lysate was bound to Glutathion-Agarose resin (3 ml resin for 5L expression culture, Serva) equilibrated in washing buffer (lysis buffer without protease inhibitors). The beads were washed extensively and protein was cleaved of the beads by overnight cleavage with 3C PreScission protease (generated in-house). The eluate was concentrated using 30 kDa molecular mass cut-off Amicon concentrators (Millipore) and applied to a Superdex 200 10/300 column (GE Healthcare) equilibrated in 50 mM Hepes, pH 8.0, 250 mM NaCl, 2 mM TCEP, 5% v/v glycerol. Relevant fractions were pooled, concentrated, flash-frozen in liquid nitrogen, and stored at −80°C. During the course of our studies, we realized that the Ndc80$^{jubaea}$ construct used for the experiments in *Figure 2A* contained a V15M mutation in Nuf2. After correcting the mutation in the Ndc80$^{jubaea}$ construct, we repeated the Ska binding assays, obtaining essentially identical results (*Figure 2—figure supplement 1*). Thus, the presence of the V15M mutation does not modify our conclusions on the ability of Ndc80$^{jubaea}$ to bind Ska. Ndc80$^{bonsai}$ and SPC24/25 were expressed and purified as described previously (*Ciferri et al., 2005*; *Ciferri et al., 2008*).

## Expression and purification of Ndc80$^{full-length}$

Expression cassettes from pLIB vectors containing NDC80, NUF2, SPC25$^{HIS}$, SPC25$^{SORT-HIS}$, SPC24, and SPC24$^{SPY}$ were combined on a pBIG1 vector using Gibson assembly as described (*Volkov et al., 2018*; *Weissmann et al., 2016*). See the Key Resources Table for the different constructs used. Baculoviruses were generated in Sf9 insect cells and used for protein expression in Tnao38 insect cells. Between 60 and 72 hr post-infection, cells were washed in PBS and stored at −80°C. All subsequent steps were performed on ice or at 4°C. Cells were thawed and resuspended in lysis buffer (50 mM Hepes, pH 8.0, 200 mM NaCl, 10% v/v glycerol, 2 mM TCEP, 20 mM imidazole, 0.5 mM PMSF, protease-inhibitor mix HP Plus (Serva)), lysed by sonication and cleared by centrifugation at 108,000 g for 60 min. The cleared lysate was filtered (0.8 μM) and applied to a 10 ml HisTrap FF (GE Healthcare) equilibrated in washing buffer (lysis buffer without protease inhibitors). The column was washed with approximately 35 column volumes of washing buffer and bound proteins were eluted with elution buffer (washing buffer containing 300 mM imidazole). Relevant fractions were pooled, diluted 5-fold with buffer A (50 mM Hepes, pH 8.0, 25 mM NaCl, 5% v/v glycerol, 1 mM EDTA, 2 mM TCEP) and applied to a 25 ml Source15Q (GE Healthcare) strong anion exchange column equilibrated in buffer A. Bound proteins were eluted with a linear gradient from 25 mM to 300 mM NaCl in 180 ml. Relevant fractions were concentrated in 50 kDa molecular mass cut-off Amicon concentrators (Millipore) and applied to a Superdex 200 16/600 or a Superose 6 10/300 column (GE Healthcare) equilibrated in 50 mM Hepes, pH 8.0, 250 mM NaCl, 2 mM TCEP (with or without 5% v/v glycerol). Size-exclusion chromatography was performed under isocratic conditions at recommended flow rates and relevant fractions were pooled, concentrated, flash-frozen in liquid nitrogen, and stored at −80°C.

## Expression and purification of Aurora B and CDK1:Cyclin-B kinases

*E. coli* BL21(DE3)-Codon-plus-RIPL cells containing the pGEX-6P-2rbs GST-Aurora B$^{45-344}$: INCENP$^{835-903}$ vector (*Girdler et al., 2008*) were grown to an OD$_{600}$ of approximately 0.4 at 25°C and protein expression was induced by the addition of 0.1 mM IPTG for ∼14 hr at 20°C. All subsequent steps were performed on ice or at 4°C. Bacterial pellets were resuspended in lysis buffer (25

mM Tris-HCl, pH 7.5, 300 mM NaCl, 1 mM TCEP, 1 mM EDTA, 1 mM PMSF), lysed by sonication, and cleared by centrifugation at 75,000 g for 30 min. Cleared lysates were applied onto a 5 ml GSH column (GE Healthcare), washed, and exposed overnight to 3C PreScission protease (generated in-house). After cleavage, Aurora B[45-344]:INCENP[835-903] was concentrated and applied to a Superdex 200 10/300 column (GE Healthcare) equilibrated in 25 mM Tris-HCl pH 7.5, 150 mM NaCl, 1 mM TCEP, 5% v/v glycerol. Relevant fractions after size-exclusion chromatography were pooled, concentrated, flash-frozen in liquid nitrogen, and stored at −80°C.

CDK1:Cyclin-B was expressed and purified as described (*Huis in 't Veld et al., 2016*). In brief, a pBIG1 vector containing codon-optimized [GST]CDK1 and [HIS]Cyclin B1[HIS] was used to express CDK1:Cyclin-B in Tnao38 insect cells. Cells were thawed and resuspended in lysis buffer (50 mM Hepes, pH 7.5, 300 mM NaCl, 5% v/v glycerol, 2 mM TCEP, a bit of lypholized DNAse, 1 mM PMSF), lysed by sonication and cleared by centrifugation. The cleared lysate was filtered (0.8 µM) and applied to a 5 ml GSH column (GE Healthcare) equilibrated in washing buffer (lysis buffer without DNAse and PMSF). The column was washed and bound proteins were eluted using washing buffer containing 20 mM glutathione. Relevant fractions were pooled, concentrated, and applied to a Superdex 200 10/300 column (GE Healthcare) equilibrated in 50 mM Hepes, pH 7.5, 300 mM NaCl, 5% v/v glycerol, 2 mM TCEP. Relevant fractions after size-exclusion chromatography were pooled, concentrated, flash-frozen in liquid nitrogen, and stored at −80°C.

## In vitro phosphorylation and fluorescent labeling

The C-terminal polyhistidine tag on SKA1 was replaced with a fluorescent peptide in an overnight labeling reaction at 10°C. Sortase, Ska, and peptide were used in an approximate molar ratio of 1:10:100. Ska complexes were exposed to Aurora B or CDK1:Cyclin-B (with ATP and MgCl$_2$) or lambda-phosphatase (with MnCl$_2$) during the Sortase labeling reaction or, in the case of pre-formed Ska:Ndc80 complexes, in a separate reaction. Phosphorylation was assessed using SDS-PAGE followed by ProQ diamond stain (Thermo Fisher), Phostag-SDS-PAGE (*Kinoshita et al., 2009*), and mass spectrometry (see below).

## Analytical size-exclusion chromatography

Proteins were mixed (exact concentrations in each experiment are indicated in the figures), incubated at 10°C for at least two hours, spun for 20 min at 13,000 rpm at 4°C, and then analyzed by size-exclusion chromatography at 4°C using an AKTAmicro system (GE Healthcare) mounted with a Superose 6 5/150 increase column (GE Healthcare) equilibrated in size-exclusion chromatography buffer (20 mM Tris-HCl pH 8, 200 mM NaCl, 2% v/v glycerol, and 2 mM TCEP) and operated at or near the recommended flow rate. Typically, fractions of 80 µL were collected and analyzed by SDS-PAGE. The absorbance of fluorescently labeled proteins was followed during chromatography using the AKTAmicro Monitor UV-900 (GE Healthcare) and by SDS-PAGE using a ChemiDoc MP system (Bio-Rad).

## SEC-MALS

SEC-MALS was performed on a Dawn Heleos II System with an Optilab T-rEX RI detector (Wyatt) and a 1260 Inifinity II LC system (Agilent). The Superose 6 increase 10/300 column (GE Healthcare) was pre-equilibrated with running buffer (50 mM HEPES pH 8.0, 200 mM NaCl, 10% Glycerol and 1 mM TCEP). Analysis was performed at room temperature with 100 µl Ska complex that was pre-diluted in running buffer from 6.7 mg/ml to 1 mg/ml.

## Analytical ultracentrifugation

Sedimentation velocity AUC was performed at 42,000 rpm at 20°C in a Beckman XL-A ultracentrifuge. Purified Ska complexes were diluted to approximately 10 µM in a buffer containing 20 mM Tris-HCl, 150 mM NaCl and 1 mM TCEP and loaded into standard double-sector centerpieces. The cells were scanned at 280 nm every minute and 500 scans were recorded for every sample. Data were analyzed using the program SEDFIT (*Schuck, 2000*) with the model of continuous c(s) distribution. The partial specific volumes of the proteins, buffer density and buffer viscosity were estimated using the program SEDNTERP. Data figures were generated using the program GUSSI.

## Crosslinking and mass spectrometry

Samples were reduced, alkylated and digested with LysC and Trypsin and prepared for mass spectrometry as previously described (*Rappsilber et al., 2007*). Obtained peptides were separated on an U3000 nanoHPLC system (Thermo Fisher Scientific). Samples were injected onto a desalting cartridge, desalted for 5 min using water in 0.1% formic acid, backflushed onto a Pepmap C18 nanoHPLC column (Thermo Fisher Scientific) and separated using a gradient from 5–30% acetonitrile with 0.1% formic acid and a flow rate of 300 nl/min. Samples were directly sprayed via a nano-electrospray source in an Orbitrap type mass spectrometer (Thermo Fisher Scientific). The mass spectrometer was operated in a data-dependent mode acquiring one survey scan and subsequently up to 15 MS/MS scans. To identify phospho-sites, the resulting raw files were processed with MaxQuant (version 1.6.1.0 or 1.6.3.4) searching for Ska and Ndc80 sequences with acetylation (N-term), oxidation (M) and phosphorylation (STY) as variable modifications and carbamidomethylation (C) as fixed modification. A false discovery rate cut off of 1% was applied at the peptide and protein levels and as well on the site decoy fraction (*Cox and Mann, 2008*).

To prepare Ska:Ndc80 for analysis by crosslinking and mass spectrometry, Ska was in vitro phosphorylated by CDK1:Cyclin-B as described above, purified using size-exclusion chromatography using a superose 6 increase 5/150 column (GE Healthcare) in crosslinking buffer (50 mM HEPES pH 8, 250 mM NaCl, 5% v/v glycerol, and 2 mM TCEP), and bound to Ndc80. Binding to Ndc80 was confirmed using analytical size-exclusion chromatography (see *Figure 2—figure supplement 3*). Approximately 100 µg of Ska, Ska:Ndc80 in crosslinking buffer were crosslinked by DSBU using an in-house generated protocol and analyzed using MeroX (v1.6.6.6 and 2.0.0.8) (*Iacobucci et al., 2018*) as described (*Pan et al., 2018*). The Ndc80 dataset reported in *Pan et al. (2018)* is deposited to the Proteome Xchange Consortium via the PRIDE partner repository (https://www.ebi.ac.uk/pride) with the data set identifier PXD010070. Proximity maps were visualized in Circos plots (*Krzywinski et al., 2009*) and arranged using Adobe Illustrator.

## Alignments

Multiple species alignments of NDC80 and SKA3 were generated using Clustal Omega (*Sievers et al., 2011*) and curated in Jalview 2 (*Waterhouse et al., 2009*).

## Low-angle metal shadowing and electron microscopy

Protein complexes were diluted 1:1 with spraying buffer (200 mM ammonium acetate and 60% glycerol) and air-sprayed onto freshly cleaved mica pieces of approximately 2 × 3 mm (V1 quality, Plano GmbH). Specimens were mounted and dried in a MED020 high-vacuum metal coater (Bal-tec). A Platinum layer of approximately 1 nm and a 7 nm Carbon support layer were evaporated subsequently onto the rotating specimen at angles of 6–7° and 45°, respectively. Pt/C replicas were released from the mica on water, captured by freshly glow-discharged 400-mesh Pd/Cu grids (Plano GmbH), and visualized using a LaB$_6$ equipped JEM-1400 transmission electron microscope (JEOL) operated at 120 kV. Images were recorded at a nominal magnification of 60,000x on a 4k × 4 k CCD camera F416 (TVIPS), resulting in 0.18 nm per pixel. Particles were manually selected using EMAN2 (*Tang et al., 2007*).

## Assembly of TS-Ndc80 modules

$T_1S_3$ and $T_3S_1$ assemblies were assembled from Traptavidin (T; addgene plasmid #26054) and Dead Streptavidin-SpyCatcher (S; addgene plasmid # 59547). Both plasmids were kind gifts from Mark Howarth (*Chivers et al., 2010*; *Fairhead et al., 2014*). $T_1S_3$-[Ndc80]$_3$ modules were prepared as described previously (*Volkov et al., 2018*). In brief, a mixture of Ndc80 (or Ndc80$^{\Delta 80}$) and $T_1S_3$ (1.8 µM) with an approximate 10-fold molar excess of Ndc80 was incubated for 12–16 hr at 10°C in the presence of protease inhibitor mix (Serva). Sortase (4 µM) and GGGGK$^{TMR}$ (137 µM) were included in the reaction to fluorescently label SPC25. In order to phosphorylate Ndc80 (19 µM), CDK1: Cyclin-B (75 nM), Aurora B (2.1 µM), ATP (1.25 mM) and MgCl$_2$ (10 mM) were also included in the overnight reaction. Reaction mixtures were applied to a Superose 6 increase 10/300 column (GE Healthcare) equilibrated in 20 mM Tris-HCl pH 8.0, 200 mM NaCl, 2% v/v glycerol, 2 mM TCEP. Relevant fractions were pooled and concentrated using 50 kDa molecular mass cut-off. Amicon

concentrators (Millipore), flash-frozen in liquid nitrogen, and stored at −80°C. Efficient phosphorylation of NDC80 was confirmed by mass spectrometry and phostag-SDS-PAGE.

## Tubulin and microtubules

Digoxigenin-labeled tubulin was produced by labeling home-purified porcine brain tubulin (*Castoldi and Popov, 2003*) according to published protocols (*Hyman, 1991*). All other tubulins were purchased from Cytoskeleton Inc. GMPCPP-stabilized seeds were made by two rounds of polymerization in 25 µM tubulin (40% dig-tubulin) supplemented with 1 mM GMPCPP (Jena Biosciences) as described (*Volkov et al., 2018*).

## TIRF microscopy

Chamber preparation and microscopy were performed as described (*Volkov et al., 2018*). In brief, both coverslips and slides were cleaned in oxygen plasma, immediately silanized, and later assembled in a chamber using double-sided tape. The chambers were first incubated with ~0.2 µM anti-DIG antibody (Roche) and passivated with 1% Pluronic F-127, followed by GMPCPP seeds (diluted 1:200 – 1:1000) and then the reaction mix. The reaction mix contained MRB80 buffer supplemented with 8 µM tubulin (4–6% labeled with HiLyte-488), 1 mM GTP, 1 mg/ml κ-casein, 0.01% methylcellulose, 4 mM DTT, 0.2 mg/ml catalase, 0.4 mg/ml glucose oxidase and 20 mM glucose; this mix was centrifuged in Beckman Airfuge for 5 min at 30 psi before adding to the chamber.

Imaging was performed at 30°C using Nikon Ti-E microscope (Nikon) with the perfect focus system (Nikon) equipped with a Plan Apo 100 × 1.45 NA TIRF oil-immersion objective (Nikon), iLas$^2$ ring TIRF module (Roper Scientific) and a Evolve 512 EMCCD camera (Roper Scientific). Images were acquired with MetaMorph 7.8 software (Molecular Devices). The final resolution was 0.16 µm/pixel. The objective was heated to 34°C by a custom-made collar coupled with a thermostat, resulting in the flow chamber being heated to 30°C. All images were analyzed using Fiji (*Schindelin et al., 2012*).

## Preparation of beads for force measurement

1 µm glass COOH-functionalized beads (Bangs Laboratories) were coated with PLL-PEG (Poly-L-lysine (20 kDa) grafted with polyethyleneglycole (2 kDa), SuSoS AG) containing 1–10% PLL-PEG-biotin, then saturated with $T_1S_3$-[Ndc80]$_3$ modules as described (*Volkov et al., 2018*). Coating density was determined for each bead preparation, and bead preparations containing on average 100–1000 Ndc80 copies per bead were used for optical trapping. This surface density was previously identified to be the optimal for efficient coupling of microtubule-generated force and rescue of shortening microtubules ends (*Volkov et al., 2018*). For experiments shown in *Figure 5* and *Figure 5—figure supplements 1* and *3*, $T_1S_3$-[Ndc80]$_3$-coated beads were additionally incubated with 400 nM Ska complex followed by extensive washing.

Flow chambers with GMPCPP-stabilized seeds were prepared as described above, the reaction mix contained MRB80 buffer with 10–12 µM tubulin, 1 mM GTP, 1 mg/ml κ-casein, 4 mM DTT, 0.2 mg/ml catalase, 0.4 mg/ml glucose oxidase and 20 mM glucose. This reaction mix was centrifuged in Beckman Airfuge for 5 min at 30 psi, and then supplemented with freshly prepared beads with or without additional 10–100 nM of Ska complex.

## Laser tweezers and experiments with the beads

DIC microscopy and laser tweezers experiments were performed as described (*Volkov et al., 2018*). Images were captured using QImaging Retiga Electro CCD cameras and MicroManager 1.4 software. At the start of each experiment, 50 frames of bead-free fields of view in the chamber were captured, averaged, and used later for on-the-fly background correction. The images were acquired at eight frames per second, subjected to background substraction and each 10 consecutive frames were averaged. The signal from the quadrant photo-detector (QPD) signal was sampled at 10 kHz without additional filtering. Experiments were performed at 0.4–0.8W of the 1064 nm laser resulting in a typical trap stiffness of 0.03–0.07 pN/nm. Trap stiffness was determined for each bead after it detached from a microtubule. If a microtubule was stalled by a bead for more than 90 s, the power of the trapping laser was increased to 4W (corresponding to a typical trap stiffness of 0.4 pN/nm)

and the bead was ruptured from the microtubule using 100 nM steps of the piezo stage. A typical force required to rupture a bead from a microtubule exceeded 40 pN.

QPD traces were analyzed in MatLab R2013b. Each force trace was analyzed manually and confirmed to correspond with a respective DIC movie in both the time of force development and the direction of microtubule pulling. We assumed that all detachment and rescue events were preceded by a stall and imposed a lower limit of 2 ms on the stall duration. The stall force was determined as a difference in mean bead displacement between the stall level and the free bead level along the microtubule multiplied by trap stiffness, corrected for the nonlinear increase of the force as a function of the distance from the trap centre (*Simmons et al., 1996*).

## Acknowledgements

MD acknowledges funding from the European Research Council Synergy Grant MODELCELL (proposal 609822). AM gratefully acknowledges funding by the Max Planck Society, the European Research Council (ERC) Advanced Investigator Grant RECEPIANCE (proposal 669686), and the DFG's Collaborative Research Centre (CRC) 1093. PJH acknowledges EMBO for the award of an EMBO short-term fellowship (grant 7203). We thank Maurits Kok and other members of the Dogterom and Musacchio laboratories for helpful discussions and comments. We are grateful to Farzad Khanipour, Caro Körner, Sara Carmignani, and the Dortmund Protein Facility for help with cloning and protein purification, to Franziska Müller, Andreas Brockmeyer, and Petra Janning for mass spectrometry, to Raphael Gasper-Schönenbrücher for SEC-MALS, to Ruben Kazner and Dongqing Pan for SV-AUC, and to Sebastiano Pasqualato (IFOM, Milan) for the NDC80^jubaea construct. We thank Jeremie Capoulade (Kavli Nanolab Imaging Center) for help with TIRF microscopy and Roland Dries for assistance with optical trapping.

## Additional information

### Funding

| Funder | Grant reference number | Author |
|---|---|---|
| European Commission | ERC AdG RECEPIANCE (proposal 669686) | Andrea Musacchio |
| Deutsche Forschungsgemeinschaft | CRC1093 | Andrea Musacchio |
| European Commission | ERC SG MODELCELL (proposal 609822) | Marileen Dogterom |
| European Molecular Biology Organization | STF7203 | Pim J Huis in 't Veld |

The funders had no role in study design, data collection and interpretation, or the decision to submit the work for publication.

### Author contributions

Pim J Huis in 't Veld, Vladimir A Volkov, Conceptualization, Validation, Investigation, Visualization, Methodology; Isabelle D Stender, Investigation, Methodology; Andrea Musacchio, Marileen Dogterom, Conceptualization, Supervision, Funding acquisition, Project administration

### Author ORCIDs

Pim J Huis in 't Veld https://orcid.org/0000-0003-0234-6390
Vladimir A Volkov http://orcid.org/0000-0002-5407-3366
Andrea Musacchio https://orcid.org/0000-0003-2362-8784
Marileen Dogterom https://orcid.org/0000-0002-8803-5261

### Decision letter and Author response

Decision letter https://doi.org/10.7554/eLife.49539.sa1
Author response https://doi.org/10.7554/eLife.49539.sa2

## Additional files

### Supplementary files

- Supplementary file 1. (**a,b**) In vitro phosphorylation of the Ska complex by CDK1:Cyclin-B (1a, concise report; 1b, full report). Phosphopeptides were detected in fractions after size-exclusion chromatography for dephosphorylated Ska + Ndc80 (*Figure 1D*; black; 1.60–1.76 ml; SDS-PAGE lanes 7 and 8), phosphorylated Ska (*Figure 1D*; light green; 1.60–1.76 ml; SDS-PAGE lanes 7 and 8), and phosphorylated Ska + Ndc80 (*Figure 1D*; dark green; 1.44–1.60 ml; SDS-PAGE lanes 5 and 6). Sites with a localization probability >0.5 and an Andromeda search engine score >100 are shown. Additional information is available in a separate table. (**c,d**) Identified crosslinks for Ska, Ndc80, and Ska:Ndc80 (1 c, concise report; 1d, full report) The unique crosslinks obtained for Ska (90 inter; 97 intra), Ndc80:Ska (86 inter; 71 intra), and Ndc80 (as in *Pan et al., 2018*) inter; 72 intra;) are reported. (**e,f**) In vitro phosphorylation of Ndc80 by Aurora B (1e, concise report; 1 f, full report) Ndc80 was Aurora B treated (or untreated) and assembled into $T_1S_3$-[Ndc80]$_3$ modules (see *Figure 4B–C*). Ndc80 phosphopeptides with a localization probability >0.5 are shown. (**g**) Table of experiments on optical tweezers reporting force measurements, duration, and outcome of stall.

- Transparent reporting form

### Data availability

All relevant data generated or analysed during this study are included in the manuscript and supporting files.

The following previously published dataset was used:

| Author(s) | Year | Dataset title | Dataset URL | Database and Identifier |
|---|---|---|---|---|
| Dongqing Pan, Tanja Bange | 2018 | Cross-linking mass spectrometry analyses of three different kinetochore protein complexes (KMN, NDC80C, MIS12C) using an MS-cleavable cross-linker, BuUrBu (DSBU) | https://www.ebi.ac.uk/pride/archive/projects/PXD010070 | Pride, PXD010070 |

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
