## [Decision Letter]

**Acceptance summary:**

This paper provides exciting new insight into how kinetochores form load-bearing attachments to dynamic spindle microtubules. At the heart of the work is the use of an in vitro reconstitution approach to reveal the minimum requirements for formation of full length Ska-Ndc80 assemblies in solution: Phosphorylation of the Ska3 subunit is the key ingredient, while the "loop" and "tail" features of the NDC80 as well as the application of force or the microtubule track are not needed. These data redefine how we interpret several in vivo findings. We were impressed with the experiments that naturally extend previous work with trivalent Ndc80 assemblies: the authors show how Ska improves the ability of Ndc80 to track shrinking microtubule ends and – importantly – lengthens force-dependent stalls and the probability of microtubule rescue. The authors reveal that stalling for more than a second increases the chance of a rescue event – the underlying mechanism and whether it is generalizable await further studies. These events happen irrespective of whether Ndc80 tail is phosphorylated raising important questions regarding tension sensing and Ska recruitment at kinetochores. Overall, this study takes an important step towards the long-term ambition to reconstitute tension-sensitive kinetochore-microtubule interface.

**Decision letter after peer review:**

Thank you for submitting your article "Molecular determinants of the Ska-Ndc80 interaction and their influence on microtubule tracking and force-coupling" for consideration by *eLife*. Your article has been reviewed by three peer reviewers, one of whom is a member of our Board of Reviewing Editors, and the evaluation has been overseen by Vivek Malhotra as the Senior Editor. The reviewers have opted to remain anonymous.

The reviewers have discussed the reviews with one another and the Reviewing Editor has drafted this decision to help you prepare a revised submission.

Summary:

This paper from the Musacchio and Dogterom labs uses a combination of in vitro biochemical, TIRF imaging and optical trap approaches to investigate the interaction between the Ska and Ndc80 complexes alone and on microtubules, with or without force. They show that the interaction between Ndc80 and Ska is regulated by Cdk1 phosphorylation of Ska3, and that this interaction in vitro does not require the Ndc80 loop or the Ndc80 tail, or microtubules. These findings represent an important advance in our understanding of the minimal requirements for assembly of tip-tracking complexes. Importantly, these in vitro findings redefine how we interpret several in vivo findings. They show that phosphorylation of trivalent Ndc80 particles and the presence of the Ndc80 tail both contribute to tracking shrinking microtubules. A key finding is that Ska increases the duration of Ndc80-mediated stalls of microtubule-depolymerisation and that this time can influence the outcome (rescue or detachment). The authors propose that this reflects a force-dependent switch from depolymerisation to polymerisation – and that Ska could regulate this switch. Overall the experiments are well done and provide important insights into our understanding of load-bearing at the kinetochore. We have the following suggestions to improve the manuscript:

Essential revisions:

1) One missing measurement is the affinity of Ndc80 for Ska in the presence and absence of phosphorylation (the bead experiments in Figure 5E show some binding of non-phosphorylated Ska to Ndc80). This would be a valuable measurement for the field. The authors should also clarify the molar ratios used in the SEC experiments: 2 to 20 μm (as stated in the Materials and methods) does not indicate whether the experiments were conducted with an excess of Ska, of Ndc80 or whether they are always in a 1:1 ratio. In this regard, the authors should comment more in the text on how they think they are probing a relevant stoichiometry and geometry for Ndc80-Ska interface in their trap assays.

2) In Figure 3 it is important to clarify what the authors can and cannot interpret from the Aurora B data. Unless the authors test pre-de/phosphorylated Ndc80 and Ska that are then mixed, the section title (“Aurora B does not affect Ska:Ndc80 binding in vitro”) and conclusion that Aurora B does not affect the binding in vitro are not appropriate given the data. The authors should either revise language or perform experiments supporting current language.

3) Tail-deleted Ndc80 was shown to have a similar residence time (i.e. affinity) but to not track depolymerizing microtubules – this is different from analysis of monovalent tail-deleted Ndc80, which shows weak affinity for microtubules. Can the similar affinity of trivalent Ndc80 with and without the tail be demonstrated using a different approach (e.g. co-sedimentation?).

4) In Figure 5 the concentrations of Ska used in the trap assay are very high, and due to the observation of hyper-stable microtubules they may be interfering with the normal features of the system. To rule out that the stabilizing effects on Ndc80 are due to lattice-binding of high Ska it is critical to repeat some of these assays with 10 nM concentration of phosphorylated and also dephosphorylated Ska. The fraction of particles in which Ska plays a role should scale with the concentration of Ska as it is titrated up. Also, what happens if Ska3 C-terminal deletion is present in solution? How would that affect interpretation of the interaction in the front part of the manuscript that is dependent on phosphorylation of the Ska3 C-terminus?

5) We believe that it is important that phosphorylated and dephosphorylated data are not pooled in order to understand the direct mechanical contribution of the Ska complex to the Ndc80-microtubule interface and what role phosphorylation plays. This will also improve the narrative from the first part of the paper (Biochemistry/phosphorylation) and the second (Biophysics) which are currently somewhat disconnected. Some expansion of the Discussion would also be helpful to provide a clearer picture of how phosphorylation impacts Ska:Ndc80 function (and how this may work in vivo).

6) We appreciate that the biophysical experiments are challenging but some increase in numbers would be important in substantiating the findings (i.e. the histograms in Figure 5—figure supplement 2 and Figure 4I and Figure 5G, H). There also needs to be statistical measurements like covariance or correlation coefficients reported for the relationship between stall duration and stall force in order to claim there are negative correlations.

7) A key finding is that Ska increases the duration of Ndc80-mediated stalls of microtubule-depolymerisation. Can the authors provide any data on whether this is specific to Ska:Ndc80 or a more general mechanisms that would occur with other linkers (i.e. a rigor kinesin)? – While not absolutely essential, we do believe this would really give the manuscript significantly more impact in the field.

8) A significant amount of biophysical analysis of Ndc80 and Ska complexes has been conducted by Asbury/Davis and Grischchuk/Cheeseman groups – the key difference here is that trivalent Ndc80 complex is used (note that antibody-based crosslinking has been shown to improve Ndc80 end tracking – e.g. in Powers et al. Figure 6). The authors need to more clearly delineate similarities and differences from the prior work and suggest reasons underlying the differences.

9) The authors need to describe the prior work in a more precise manner and be more cautious in their statements about what their work does versus does not imply. In particular, they cannot make strong claims about the degree to which the interaction they focus on in vitro accounts for Ska localization dynamics in vivo. The authors state that prior work has indicated a requirement for the Ndc80 tail in Ska localization – this is not correct. For example, the Cheerambathur manuscript cited in support of this claim showed that Ska localized in tail-deleted Ndc80 (see Figure 3E and F in that paper). The authors should more clearly acknowledge Zhang et al., 2017, in the Introduction for the discovery that phosphorylation of Ska3 by Cdk1 is important for Ska kinetochore localization and for interaction with Ndc80.

---

## [Author Response]

Essential revisions:1) One missing measurement is the affinity of Ndc80 for Ska in the presence and absence of phosphorylation (the bead experiments in Figure 5E show some binding of non-phosphorylated Ska to Ndc80). This would be a valuable measurement for the field. The authors should also clarify the molar ratios used in the SEC experiments: 2 to 20 μm (as stated in the Materials and methods) does not indicate whether the experiments were conducted with an excess of Ska, of Ndc80 or whether they are always in a 1:1 ratio. In this regard, the authors should comment more in the text on how they think they are probing a relevant stoichiometry and geometry for Ndc80-Ska interface in their trap assays.

We have now added the concentrations of Ndc80 and Ska to all SEC experiments in the main and supplementary figures. In brief, a two-fold molar excess of Ndc80 was often used to clearly monitor Ska’s changed retention on the column due to Ndc80 binding. Typical concentrations were e.g. 4 μm for Ska and 8 μm for Ndc80 (see Figure 1D-E). Lower Ndc80 concentration (Ska 4 μm and Ndc80 2 uM) reduced Ndc80:Ska binding to levels that prevented binding assessment by gel filtration (Author response image 1). Our conclusion from these experiments is that phosphorylated Ska binds to Ndc80 at low μM concentrations and that the binding is not complete when Ndc80 and Ska are present at equimolar concentrations.

It is important to point out that the situation is very different in our single-molecule TIRF microscopy and optical trap experiments. In those setups we use a) Ska complexes at much lower concentrations (100 pM – 100 nM), b) multivalent Ndc80 complexes, and c) microtubules that provide a docking site for both Ska and Ndc80. The complexity of this setup allowed us to see long co-localization between phosphorylated Ska and multivalent Ndc80 on microtubules when Ska was used at concentrations as low as 100 pM. The presence of many binding interfaces prevented us to approximate a K*d* between Ndc80 and Ska. The reviewers are right to point out that under these conditions, the phosphorylation of Ska3 contributes to Ndc80 binding but is no longer strictly required. We have revised our text to highlight these observations and to explain how our experimental setup mimics crucial properties of a human outer kinetochore (changes throughout the text, in particular in the subsection “Ska stabilizes end-on Ndc80-microtubule interactions under force” and in the Discussion).

We agree with the reviewers that accurately determined binding affinities between Ndc80 and phosphorylated/non-phosphorylated Ska would be of value for the field. The binding affinity can be guessed, but not precisely determined, from our non-equilibrium SEC experiments. In the past months, we therefore attempted to determine Ndc80:Ska binding affinities by equilibrium experiments, including isothermal titration calorimetry (ITC) and fluorescence polarization (FP). We succeeded in the purification of large amounts of these multiprotein complexes expressed in insect cells and with the right phosphorylation state, which was challenging by itself. Regretfully, however, our efforts were not successful, and for different reasons. An apparent tendency of Ska to multimerize at higher μM concentrations (consistent with SEC-MALS analysis in Figure 1—figure supplement 1) hindered ITC measurements due to the generation of very large dilution heats. Conversely, fluorophores positioned at the C-terminal end of Ska1 (as used throughout the manuscript) retained full fluorescence polarization in the absence of Ndc80, preventing the inspection of Ndc80:Ska binding through changes in polarization. This is not unexpected given the large size and the elongated nature of these complexes. So, regretfully, we are currently unable to provide a number for the binding affinity of the Ndc80-Ska complex.

We also attempted to use fluorescence polarization to determine the binding between Ndc80 and a fluorescently labeled peptide that encompasses the sequence of the Ska3 binding site around residues 358 and 360. For this purpose we ordered three different peptides with sequences encompassing different segments of SKA3 (351-367, 334-367, and 351-377) in a double phosphorylated (phospho-T358 and phospho-T360) or non-phosphorylated form. To our satisfaction, the phosphorylated Ska3 351-377 peptide bound better to Ndc80 than the other five. This demonstrates that a minimal fragment also relies on the phosphorylation of T358 and T360 for the binding to Ndc80 and hints towards a potentially interesting contribution from the Ska3 368-377 stretch. This stretch is predicted to contain an α-helix and further experiments will be required to determine its importance. Despite these results, we have not been able to saturate polarization despite efforts to concentrate Ndc80^full-length^ or Ndc80^jubaea^ to (unprecedentedly high) concentrations exceeding 100 uM. This prevented us from affinity determination. Moreover, SEC experiments demonstrated efficient Ndc80:Ska binding at low μM concentrations whereas the polarization signal of the phospho 351-377 Ska3 peptide increases only minimally when Ndc80 is present at such concentrations. Full-length phosphorylated Ska thus binds Ndc80 with higher affinity than the phosphopeptide. Whether this is due to Ska’s ability to dimerize or multimerize remains to be tested.

We have thus attempted to determine the binding between Ndc80 and phosphorylated Ska using different methods. Quantitative assessment proved difficult due to the size of these complexes, the non-monomeric nature of Ska at higher concentrations, and the dependency on the right Ska3 phosphorylation pattern. Our SEC experiments indicate phosphorylation-dependent binding at low μM concentrations, EM micrographs indicate that Ndc80 recruits dimers of phosphorylated Ska, and our single-molecule and single-molecule experiments indicate that multivalent Ndc80 recruits phosphorylated Ska to microtubules at concentrations as low as 0.1 nM. As presented in Figure 5—figure supplement 1, the incubation of beads coated with trivalent Ndc80 with varying concentrations of Ska indicate the phosphorylation of SKA3 increases Ska’s affinity for Ndc80 in a limited concentration range (around 10 nM), while at higher concentrations even dephosphorylated Ska is bound to the Ndc80 oligomers. This is consistent with data in Figure 5C: Ska enhances the end-tracking of tailless Ndc80 trimers in a concentration-dependent manner and the phosphorylation-dependency of this is most pronounced at a Ska concentration of 10 nM.

2) In Figure 3 it is important to clarify what the authors can and cannot interpret from the Aurora B data. Unless the authors test pre-de/phosphorylated Ndc80 and Ska that are then mixed, the section title (“Aurora B does not affect Ska:Ndc80 binding in vitro”) and conclusion that Aurora B does not affect the binding in vitro are not appropriate given the data. The authors should either revise language or perform experiments supporting current language.

We agree with the reviewers that our description of this experiment was imprecise. Aurora B was indeed added to a pre-formed Ska:Ndc80 complex and we did thus not assess effects of Aurora B on the formation of this complex. We have revised the text and now refer consistently to this experiment as: “Aurora B does not disrupt a pre-formed Ska:Ndc80 complex”.

3) Tail-deleted Ndc80 was shown to have a similar residence time (i.e. affinity) but to not track depolymerizing microtubules – this is different from analysis of monovalent tail-deleted Ndc80, which shows weak affinity for microtubules. Can the similar affinity of trivalent Ndc80 with and without the tail be demonstrated using a different approach (e.g. co-sedimentation?).

We agree with the reviewers that unchanged residence time of tailless Ndc80 trimers is surprising given that monomeric Ndc80^∆80^ was previously reported to bind more weakly to the microtubules than the full-length complex. We confirmed using co-sedimentation assays that our tailless monomeric Ndc80^∆80^ bound microtubules less effectively than wild type (Author response image 2). This is consistent with previous literature. Although we observed that wild-type Ndc80 trimers as well as tailless Ndc80 trimers sediment almost completely in the presence of microtubules, a quantitative assessment of binding was unfortunately prevented by the sedimentation of trivalent Ndc80 in the absence of microtubules. We conclude that Ndc80 trimers on a streptavidin-based scaffold (spanning >100 nm and ~0.5 MDa) are not suited for this assay.

**Author response image 2. respfig2:** 

4) In Figure 5 the concentrations of Ska used in the trap assay are very high, and due to the observation of hyper-stable microtubules they may be interfering with the normal features of the system. To rule out that the stabilizing effects on Ndc80 are due to lattice-binding of high Ska it is critical to repeat some of these assays with 10 nM concentration of phosphorylated and also dephosphorylated Ska. The fraction of particles in which Ska plays a role should scale with the concentration of Ska as it is titrated up. Also, what happens if Ska3 C-terminal deletion is present in solution? How would that affect interpretation of the interaction in the front part of the manuscript that is dependent on phosphorylation of the Ska3 C-terminus?

We agree with the reviewers that some stabilization of microtubule lattice is expected at 100 nM Ska and have followed the reviewer’s suggestion to repeat experiments in the optical trap with 10 nM phosphorylated and dephosphorylated Ska. We have done so for all three types of Ndc80 trimers (non-phosphorylated, Aurora B-treated, and tailless) and are pleased to report 93 new measurements in Figure 5—figure supplements 2 and 3. These results are also included in our summary in Figure 6.

Compared to experiments in the absence of Ska, we note that 10 nM Ska results in an increase in the fraction of rescues after stalled microtubule depolymerization for phosphorylated and tailless Ndc80 trimers, and in a general increase in the stall duration in all conditions. Although the addition of 10 nM and 100 nM appears to have a comparable effect, we cannot conclude this with certainty with the current number of measurements.

It is interesting to point out that we did observe three ‘superstall’ events in the presence of Ska lacking SKA3^∆C^. We hypothesize that this hyper-stabilization of microtubule ends requires dephosphorylated Ndc80 tails and the accumulation of Ska at microtubule ends. Apart from these three events, Ska lacking SKA3^∆C^ did not result in the increase in the force or duration of Ndc80-mediated microtubule stalls (Figure 5—figure supplement 3D) and did not bind to beads coated with Ndc80 trimers (Figure 5D). Since also the addition of Ska lacking SKA1^MTBD^ (also in Figure 5—figure supplement 3D) did not alter force-coupling, we conclude that both SKA3:Ndc80 and SKA1:microtubule interactions are prerequisites for the stabilization of Ndc80-mediated microtubule stalls. We have improved our explanation of these phenomena throughout the text, see e.g. –subsection “Ska stabilizes end-on Ndc80-microtubule interactions under force”.

5) We believe that it is important that phosphorylated and dephosphorylated data are not pooled in order to understand the direct mechanical contribution of the Ska complex to the Ndc80-microtubule interface and what role phosphorylation plays. This will also improve the narrative from the first part of the paper (Biochemistry/phosphorylation) and the second (Biophysics) which are currently somewhat disconnected. Some expansion of the Discussion would also be helpful to provide a clearer picture of how phosphorylation impacts Ska:Ndc80 function (and how this may work in vivo).

We have included separate force-duration graphs for phosphorylated and dephosphorylated Ska at 10 and 100 nM for three types of Ndc80 trimers in Figure 5—figure supplement 2, and also relevant distributions in Figure 5—figure supplement 3. The overlapping and scattered data prevents us to determine the effect of Ska phosphorylation under these conditions. We have therefore presented pooled data in Figures 5 and 6.

As described above, we attribute the reduced importance of Ska phosphorylation for Ska:Ndc80 binding in our biophysical experiments to the multivalent organization of Ndc80 in our setup and to the presence of microtubules that provide a docking site for both Ska and Ndc80. At this point it is important to point out that we have previously demonstrated that monomeric Ndc80 -also at high local concentrations- is quite inefficient as a microtubule force-coupler (Volkov et al., 2018, Figure 4D). As described above, we have updated the Results and the Discussion sections of our manuscript to highlight these differences and to connect the two parts of the paper.

6) We appreciate that the biophysical experiments are challenging but some increase in numbers would be important in substantiating the findings (i.e. the histograms in Figure 5—figure supplement 2 and Figure 4I and Figure 5G, H). There also needs to be statistical measurements like covariance or correlation coefficients reported for the relationship between stall duration and stall force in order to claim there are negative correlations.

Following the reviewer’s suggestion, we have repeated force-coupling experiments in many different conditions to increase our number of measurements. The updated data are presented in Figures 4I, 5H, and Figure 5—figure supplements 2 and 3. We also calculated Spearman correlations for these force vs. stall data (Author response table 1).

**Author response table 1. resptable1:** 

	no Ska	+ 100 nM Ska
T_1_S_3_-[Ndc80]_3_	0.35 (*p = 0.009*)	0.42 (*p = 0.014*)
T_1_S_3_-[Ndc80-P]_3_	-0.23 (*p = 0.23*)	0.27 (*p = 0.03*)
T1S3-[Ndc80^∆80^]3	-0.16 (*p = 0.45*)	0.47 (*p = 0.003*)

While negative correlation coefficients in the absence of Ska proved to be insignificant, addition of Ska converts them into significant positive correlations. These correlation coefficients, however, should be treated with caution as they describe data with two outcomes (detachment or rescue) that have very different stall durations. Stalls that are followed by a rescue last roughly an order of magnitude longer. Therefore, the positive correlations could be attributed to the (increasing) appearance of rescues in the presence of Ska. We think that the fraction of rescue events, as reported in Figure 6, is a straightforward and meaningful to look at these data.

7) A key finding is that Ska increases the duration of Ndc80-mediated stalls of microtubule-depolymerisation. Can the authors provide any data on whether this is specific to Ska:Ndc80 or a more general mechanisms that would occur with other linkers (i.e. a rigor kinesin)? – While not absolutely essential, we do believe this would really give the manuscript significantly more impact in the field.

We thank the reviewers for this interesting suggestion. We believe that the correlation of the probability to rescue with the duration of the stall is a property of a microtubule end, because very different sets of conditions we have tested so far resulted in practically the same curve (see Figure 6C-D). We suspect that any molecule able to keep a microtubule end in a stalled conformation for a long enough time will produce a similar relationship. However, to our knowledge a rigor kinesin has not been shown to stall microtubule shortening. This said, while we see that this is a very interesting question, which may eventually reveal a fundamental aspect of the behavior of force coupling, we note that the analysis of different linker would require the same inquisitive approach applied here (e.g. using various mutants, etc.) and we feel that this is a long-term question that, despite its interest, is outside the scope of this manuscript.

8) A significant amount of biophysical analysis of Ndc80 and Ska complexes has been conducted by Asbury/Davis and Grischchuk/Cheeseman groups – the key difference here is that trivalent Ndc80 complex is used (note that antibody-based crosslinking has been shown to improve Ndc80 end tracking – e.g. in Powers et al. Figure 6). The authors need to more clearly delineate similarities and differences from the prior work and suggest reasons underlying the differences.

We thank the reviewers for this suggestion. We have emphasized these differences throughout our revised Discussion. In summary, key novel features of our study include:

1) Systematic analysis of molecular determinants for the interaction of purified recombinant full-length Ska and Ndc80 complexes. Previous work was performed with protein fragments (Schmidt et al., 2012, Chakraborty et al., 2019), in the absence of Ska phosphorylation (Schmidt et al., 2012, Chakraborty et al., 2019, Helgeson et al., 2018), and using cell lysates (Zhang et al., 2017). This allowed us to distinguish direct Ska-Ndc80 binding (phosphorylated SKA3^C^:NDC80/NUF2 coiled coil) from molecular features that are important for kinetochore-microtubule interactions (Ndc80 tail, Ndc80 loop, SKA1^MTBD^).

2) Controlled oligomerization of Ndc80, which we have previously found crucial for physiologically relevant interactions with microtubule ends (e.g. stalling and rescue of microtubule shortening), but which we also now report as important for efficient loading of Ska. Previous biophysical studies were performed with either monomeric Ndc80 (Schmidt et al., 2012, Chakraborty et al., 2019, Helgeson et al., 2018), or poorly controlled multimers of Ndc80 without Ska (McIntosh et al., 2008; Powers et al., 2009).

3) In our assays we let the microtubule develop the force, mimicking kinetochore-microtubule attachments in the cell. This allows us to study the efficiency of microtubule force production, and the parameters of the kinetochore-mediated stalls. Previous studies were performed either in the absence of external force (Schmidt et al., 2012, Chakraborty et al., 2019), or under conditions where force was applied to rupture the bead-microtubule connection (Helgeson et al., 2018), which limited the analysis of the effect of Ska on the stability of the stalled microtubule conformation, which we find to be crucial.

9) The authors need to describe the prior work in a more precise manner and be more cautious in their statements about what their work does versus does not imply. In particular, they cannot make strong claims about the degree to which the interaction they focus on in vitro accounts for Ska localization dynamics in vivo. The authors state that prior work has indicated a requirement for the Ndc80 tail in Ska localization – this is not correct. For example, the Cheerambathur manuscript cited in support of this claim showed that Ska localized in tail-deleted Ndc80 (see Figure 3E and F in that paper). The authors should more clearly acknowledge Zhang et al., 2017, in the Introduction for the discovery that phosphorylation of Ska3 by Cdk1 is important for Ska kinetochore localization and for interaction with Ndc80.

We thank the reviewers for pointing this out and agree that the precise wording is crucial for the description of the recruitment of Ska to kinetochores. in vitro studies (like our current study) lack the complexity of interactions observed in vivo. At the same time, this allowed us to distinguish direct Ska-Ndc80 binding (phosphorylated SKA3^C^:NDC80/NUF2 coiled coil) from molecular features that are important for kinetochore-microtubule interactions (Ndc80 tail, Ndc80 loop, SKA1^MTBD^). Our experiments show that the Ndc80 tail does not interact with Ska.

Janczyk et al., 2017 reported that Ndc80-tail directly recruits Ska to kinetochores, and therefore we respectfully disagree with the reviewer’s concern that this interaction was not suggested in the prior literature. However, we agree with the reviewers’ interpretation of the data in Cheerambathur et al., 2017. We cite this paper at different points in our manuscript and write e.g.: “In another study, the Ndc80 N-terminal tail was shown to regulate the localization of Ska to kinetochores (Cheerambathur et al., 2017)”, which does not imply a direct interaction and is in agreement with major conclusions from that study.

We have updated our reference to the Zhang et al. 2017 study in the Introduction and changed “In yet another study…” to “In a key recent study …”. This change reflects how important this work was for our current study (as is also reflected by our frequent references to this work throughout our manuscript).